# Structural determinants of lipid specificity within Ups/PRELI lipid transfer proteins

Xeni Miliara[1], Takashi Tatsuta[2,3], Jamie-Lee Berry[1], Sarah L. Rouse[1], Kübra Solak[2,3], Dror S. Chorev[4], Di Wu[4], Carol V. Robinson[4], Stephen Matthews [1] & Thomas Langer [2,3]

Conserved lipid transfer proteins of the Ups/PRELI family regulate lipid accumulation in mitochondria by shuttling phospholipids in a lipid-specific manner across the intermembrane space. Here, we combine structural analysis, unbiased genetic approaches in yeast and molecular dynamics simulations to unravel determinants of lipid specificity within the conserved Ups/PRELI family. We present structures of human PRELID1–TRIAP1 and PRELID3b–TRIAP1 complexes, which exert lipid transfer activity for phosphatidic acid and phosphatidylserine, respectively. Reverse yeast genetic screens identify critical amino acid exchanges that broaden and swap their lipid specificities. We find that amino acids involved in head group recognition and the hydrophobicity of flexible loops regulate lipid entry into the binding cavity. Molecular dynamics simulations reveal different membrane orientations of PRELID1 and PRELID3b during the stepwise release of lipids. Our experiments thus define the structural determinants of lipid specificity and the dynamics of lipid interactions by Ups/PRELI proteins.

---

[1] Department of Life Sciences, Imperial College London, Sir Ernst Chain Building, South Kensington, London SW7 2AZ, UK. [2] Max-Planck-Institute for Biology of Ageing, Joseph-Stelzmann-Str. 9b, 50931 Cologne, Germany. [3] Cologne Excellence Cluster on Cellular Stress Responses in Aging-Associated Diseases (CECAD), Center for Molecular Medicine (CMMC), University of Cologne, Joseph-Stelzmann-Str. 26, 50931 Cologne, Germany. [4] Physical and Theoretical Chemistry Laboratory, Department of Chemistry, University of Oxford, South Parks Road, Oxford OX1 3TA, UK. These authors contributed equally: Xeni Miliara, Takashi Tatsuta, Jamie-Lee Berry. Correspondence and requests for materials should be addressed to S.M. (email: s.j.matthews@imperial.ac.uk) or to T.L. (email: Langer@age.mpg.de)

Mitochondria are dynamic organelles involved in coordinating a plethora of cellular processes in both health and disease. Proper mitochondrial function requires a highly coordinated system of proteins and phospholipid supply. Phospholipids and their precursors must be delivered to mitochondria, shuttled between membrane leaflets and transported across the mitochondrial intermembrane space (IMS)[1,2]. In particular, the accumulation of the mitochondria-specific phospholipid cardiolipin (CL), as well as phosphatidylethanolamine (PE), which are synthesised within mitochondria, is necessary for normal cell function. Mitochondrial CL and PE deficiency results in abnormal mitochondrial morphology and is associated with embryonic lethality in mice[3–6]. The cellular consequences are typically altered respiratory pathways and protein import as well as oxidative stress leading to apoptotic cell death[7–10].

The first molecular details of phospholipid transport to and within mitochondria were recently uncovered with the identification of the Ups–Mdm35 family in yeast and the homologous PRELI–TRIAP1 system in humans[11–14]. TRIAP1 (yeast Mdm35) and PRELI (yeast Ups) proteins reside within the IMS where they regulate phospholipid metabolism. Homologues of the Ups/PRELI protein family (PRELID1, PRELID3a, PRELID3b in humans and Ups1, Ups2 and Ups3 in yeast) form complexes with TRIAP1/Mdm35 and together facilitate intramitochondrial transport in a lipid-specific manner. Specific phospholipids are extracted from the mitochondrial outer membrane (OM), carried across the IMS and inserted into the mitochondrial inner membrane (IM), towards the synthesis machineries of CL and PE[2]. It has been established that the Ups1–Mdm35 complex directly transfers phosphatidic acid (PA) between mitochondrial membranes allowing CL synthesis in the IM[13]. Similarly, the human homologue PRELID1–TRIAP1 mediate PA transfer within the cardiolipin synthetic pathway[14]. More recently, Ups2–Mdm35 was shown to mediate the transfer of phosphatidylserine (PS) from donor to acceptor liposomes, in a similar manner to the Ups1–Mdm35[15,16]. PRELID3b (previously known as SLMO2) is the human homologue of Ups2 and can functionally replace Ups2 when expressed in ups2Δ yeast cells[15]. Thus, lipid transfer proteins of the Ups/PRELI-family shuttle phospholipids across the IMS in a phospholipid specific manner. On the other hand, the distribution of phosphatidylcholine (PC), the most abundant phospholipid in mammalian cells, within mitochondrial membranes is controlled by the steroidogenic acute regulatory protein (StAR)-related lipid transfer (START) protein, StarD7[17,18].

Detailed mechanistic insight into the PRELID–TRIAP1 and Ups–Mdm35 systems came with the high-resolution crystal structure determination of human PRELID3a–TRIAP1 and yeast homologue Ups1–Mdm35[19–21], alongside mutagenesis and the characterisation of their transfer activities. The architecture of the PRELI domain is reminiscent of other lipid transfer proteins such as the START family[22], including the phosphoinositide transfer (PITP) and ceramide transfer proteins (CERT). These are characterised by an internal lipid binding pocket, surrounded by a β-sheet and α-helices with several flexible loops with putative regulatory roles. The partner protein TRIAP1/Mdm35 has an important role in mitochondrial import and folding of the PRELI domain as well modulating its membrane affinity[13,19,20,23,24].

Although these studies illuminated key features important for phospholipid transfer, the structural determinants of phospholipid specificity of different members of this lipid transfer protein family remain unknown. Here, we combine crystal structures with reverse genetics in yeast and molecular simulations, to unravel the structural determinants of the lipid specificity within Ups/PRELI-like proteins.

## Results

### Crystal structures of PRELID1–TRIAP1 and PRELID3b–TRIAP1.

We first determined the crystal structures for human PRELID1 and PRELID3b in complex with TRIAP1. A variety of expression constructs were explored to supply suitable reagents for crystallography, including histidine tags on either TRIAP1 or the PRELI domain for purification and a maltose binding protein fusion (MBP) for crystallisation. A construct of PRELID1 in which an unstructured region from residue 173 was omitted and Cys112 and 115 were substituted for serine residues facilitated the best soluble protein production (PRELID1$^{1–173}$) with no effect on activity. PRELID1$^{1–173}$–TRIAP1 and PRELID3b–TRIAP1 yielded crystals that diffracted to atomic resolution. The structures of PRELID1$^{1–173}$–TRIAP1 and PRELID3b–TRIAP1 were solved by molecular replacement and refined to 1.98 and 2.91 Å resolution, respectively. The overall architecture of both the PRELID1$^{1–173}$ and the PRELID3b domain is consistent with the previously solved structures of Ups1[20,21] and PRELID3a[19] (Fig. 1). The PRELI domain comprises a seven β-stranded antiparallel concaved β-sheet packed against three α helices, which produces a tunnel-like cavity within the hydrophobic core capped by the Ω loop (L4-α2-L5 loop).

Both PRELID1$^{1–173}$ and PRELID3b in complex with TRIAP1 form swapped dimers under the crystallographic conditions by exchanging the α3 C-terminal helix between two symmetry related molecules (Supplementary Figure 1). Although a similar swapped dimeric structure has also been reported for Ups1[20] and was attributed to crystallisation artefacts, a functional role for dimerisation has not been ruled out. The relative domain orientations are different when compared to each other, which is due to conformational differences in the α3 C-terminal helix and the loop connecting this to the β-sheet. On one extreme, large movements of the Ω loop and the C-terminal α3 helix away from the core of PRELID3b widens the entrance leading to the inner cavity of the PRELI domain (Fig. 1c). Furthermore, higher crystallographic B factors observed for the Ω loop also reflects enhanced mobility for this region. In contrast for PRELID1$^{1–173}$, the Ω loop and the C-terminal α3 helix are much more closed than PRELID3b (Fig. 1c). These differences are likely to be important for the lipid specificity of these lipid transfer proteins.

### Swapping lipid specificity using reverse genetics in yeast.

To identify the structural determinants of lipid specificity within the Ups/PRELI family of lipid transfer proteins, we turned our attention to yeast orthologues of this family. Ups/PRELI proteins of yeast and human share the same structural fold and can functionally substitute for each other[6,15]. We therefore used an unbiased genetic screen in yeast to identify amino acids critical for the substrate specificity of Ups/PRELI proteins. The loss of the PRELID3b-homologue Ups2 is lethal in yeast cells lacking Phb1 or Phb2[11], subunits of prohibitin membrane scaffolds in the IM[25]. Cell growth can be restored by expression of Ups2, but only to a limited extent by the expression of the PRELID1 homologue Ups1 (Fig. 2c), reflecting their different specificities for PS and PA. We therefore performed a reverse genetic screen for Ups1 variants that allow survival of ups2Δphb1Δ yeast cells (Fig. 2a). Variants of UPS1 were generated by random PCR mutagenesis and expressed in ups2Δphb1Δ cells complemented with plasmid-borne Phb1. After plasmid-shuffling using 5-fluorouracil-6-carboxylic acid (5′FOA) to exclude the Phb1-expressing plasmid, we isolated and sequenced UPS1 variants that allowed growth of ups2Δphb1Δ cells. We identified 332 amino acid exchanges in 121 sequenced UPS1 genes (2.74 mutations per clone, Supplementary Data 1). Mutations affected most frequently amino acids K58, T76, T95, E108, F133 and M135 of Ups1, which

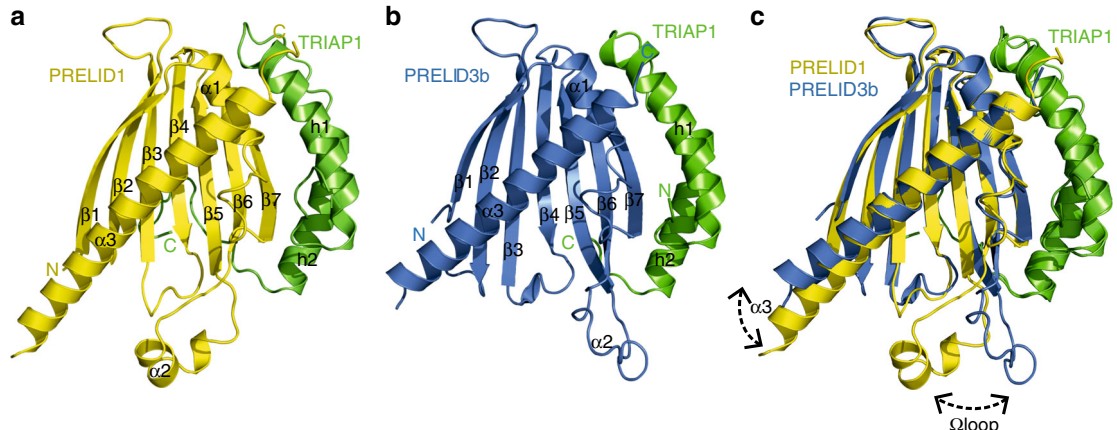

**Fig. 1** Crystal structures of PRELID1[1–173] and PRELID3b in complex with TRIAP1. **a** Ribbon cartoon of PRELID1[1–173] (yellow) in complex with TRIAP1 (green) generated in pymol graphics package. Secondary structure elements are shown in black, and indicated as; α = α-helix, β = β-strand for PRELID1[1–173] and h = helix for TRIAP1, and numbered accordingly from N-terminus to the C-terminus. N and C indicate termini that are coloured according to protein chain. **b** Ribbon cartoon of PRELID3b (blue) in complex with TRIAP1 (green) generated in pymol graphics package. Secondary structure elements are shown in black, and indicated as; α = α-helix, β = strand for PRELID3b and h = helix for TRIAP1 and numbered accordingly from N-terminus to the C-terminus. N and C indicate termini that coloured according to protein chain. **c** Comparison of PRELID1[1–173]-TRIAP1 (yellow-green) and PRELID3b–TRIAP1 complexes (blue-green). Smooth ribbon representation of the structural alignment of PRELID1[1–173]-TRIAP1 and PRELID3b–TRIAP1 complexes scoring RMSD = 1.3 Å over the backbone atoms of 223 residues. Secondary structure elements (α = α-helix, β = β-strand) presenting differences between the two models are indicated. The position of the flexible α2 and C-terminal α3 is indicated with dotted arrows

either flank the Ω loop, are located at the beginning of the C-terminal helix (α3) or in the β-sheet of the PRELI domain (Fig. 2b, Supplementary Figure 2A, Supplementary Table 1).

It is conceivable that a combination of mutations in Ups1 is required to alter its lipid specificity and to allow cell growth when expressed in *ups2Δphb1Δ* cells. Alternatively, Ups1 variants harbouring only individual mutations may be able to support growth of *ups2Δphb1Δ* cells. To distinguish between these possibilities, we mutated the most frequently affected amino acid residues of Ups1 by site-directed mutagenesis and expressed the obtained Ups1 variants in *ups2Δphb1Δ* cells (Fig. 2c). The growth of *ups2Δphb1Δ* cells was significantly improved upon expression of the mutant forms of Ups1 when compared to cells expressing wildtype Ups1, demonstrating that the replacement of single amino acids in Ups1 is sufficient to promote cell growth (Fig. 2c). The ability of the Ups1 variants to substitute for the loss of Ups2 suggests that the mutations enabled Ups1 to transport PS, pointing to a critical role of specific residues flanking flexible loops and in the β-sheet of the PRELI domain of Ups1 for its lipid specificity.

To examine whether the mutation of these residues affects the ability of Ups1 for PA transfer, we expressed Ups1 variants in *ups1Δ* cells and assessed cell growth (Supplementary Figure 2B). Similar to Ups1, the Ups1 variants restored normal growth of *ups1Δ* cells, indicating that these mutants retain PA transfer activity. We therefore conclude that the amino acids K58, T76, T95, E108, F133 and M135 have a pivotal role in substrate selection. Mutations in these residues appear to allow PS transport by Ups1 without inhibiting its ability to transport PA.

**Lipid transfer activities of Ups1 variants**. To analyse directly phospholipid transport by the Ups1 variants, we expressed them together with Mdm35 in a cell-free protein expression system in vitro and purified soluble heterodimeric complexes to near homogeneity (Supplementary Figure 3A). Lipid transport was quantitatively assessed in vitro monitoring the dequenching of NBD-labelled lipids upon transfer to acceptor liposomes lacking fluorescent lipids[14]. Consistent with previous reports[13], Ups1–Mdm35 complexes transferred PS only inefficiently when

compared to PA (~7.6% of PA transfer activity; Fig. 3a, b, Supplementary Figure 3D). The mutations K58V or E108D in Ups1 increased PS transport by ~200% (Fig. 3a, c). Although to a lesser extent, other Ups1 variants also showed increased PS transfer activity when compared to wildtype Ups1–Mdm35 complexes (Fig. 3a, c, Supplementary Figure 3D). Whereas mutations of the amino acids K58 and E108 of Ups1 and, to some extent, mutations of T76, T95 and M135 of Ups1 enable PS transfer, the mutations decreased the ability of Ups1 to transfer PA, with K58V and E108D showing the most pronounced effects (Fig. 3b, c, Supplementary Figure 3D). Thus, the relative PS/PA transfer activity of the Ups1 variants identified in the reverse genetic screen is significantly increased (Fig. 3d).

Although frequently identified in our reverse genetic screen, mutations in the amino acids T76, T95 and M135 had only a moderate effect on the PS transfer activity of Ups1. To gain further insight into the roles of these residues, we introduced the corresponding mutations and E108D in the Ups1 variant harbouring K58V. The mutations substantially increased PS transfer by Ups1-K58V, while the ability for PA transport remained largely unaffected, indicating that these residues limit PS transfer by Ups1-K58V (Fig. 3c, d, Supplementary Figure 3B, 3C, 3D). We also monitored PA/PS transfer by Ups1, Ups1-K58V and Ups1-K58VE108D using donor liposomes containing both non-fluorescent PS and PA (Supplementary Figure 3E, 3F). Whereas Ups1 transfers PA in a highly specific manner, Ups1-K58V and Ups1-K58VE108D transfer PS and PA, confirming their increased ability to transfer PS.

Together, these results establish the identified amino acid residues, in particular K58, as critical determinants for the lipid specificity of Ups1. Notably, the elevated PS transfer activity of various Ups1 mutants enables them to substitute for Ups2 in vivo. Expression of the Ups1 variants K58V, E108D or K58VE108D in *phb1Δups2Δ* cells significantly increased mitochondrial PE levels (Supplementary Figure 2C, D).

**K58 is a conserved determinant of PS specificity**. Sequence alignments of the Ups/PRELI-family of lipid transfer proteins revealed that K58 is highly conserved among members of the

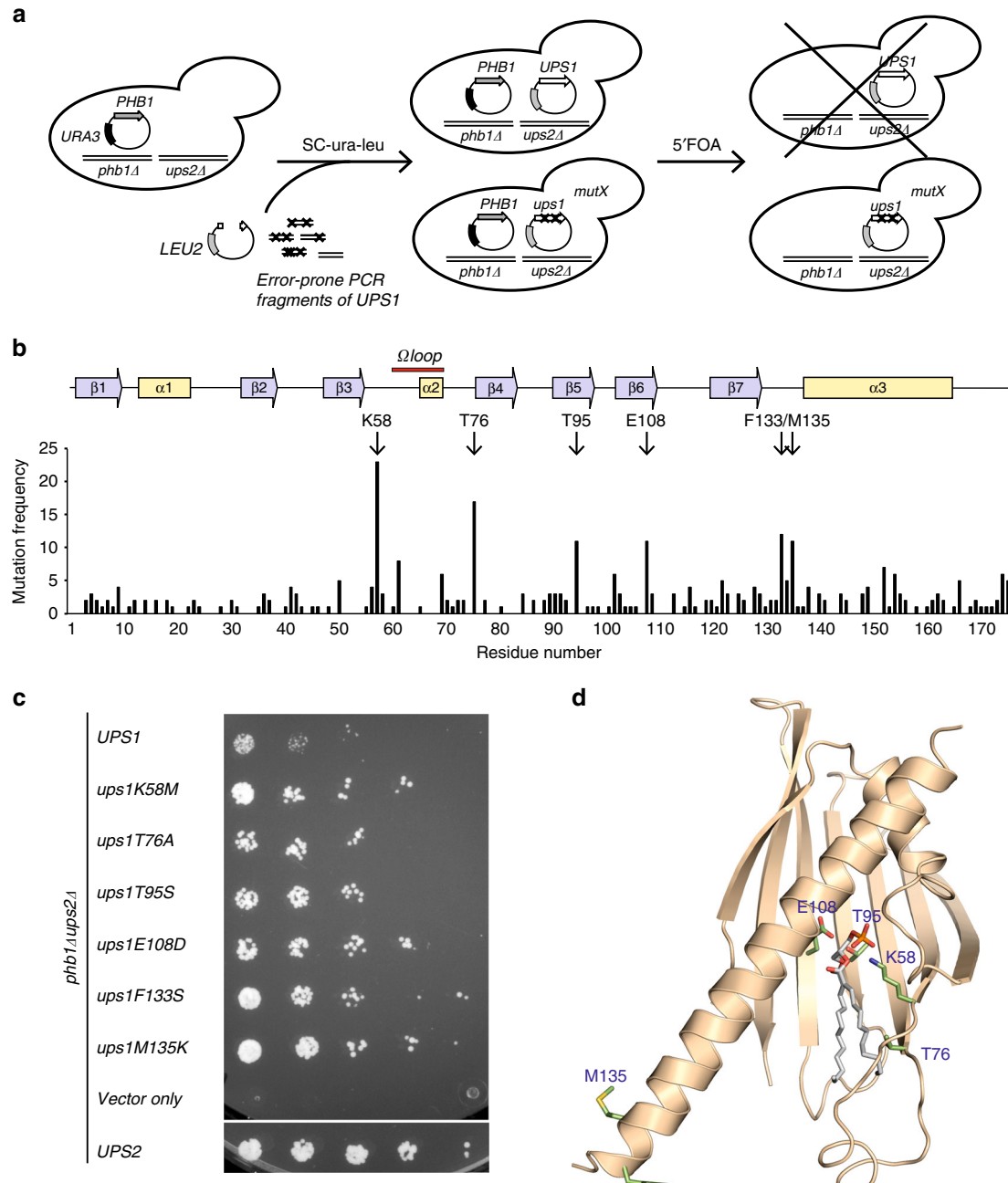

**Fig. 2** Identification of Ups1 variants transferring PS. **a** Reverse genetic screen for Ups1 variants substituting for Ups2. *ups2Δphb1Δ* cells harbouring plasmid-borne *PHB1* and *URA3* were transformed with a linearised yeast plasmid and with PCR fragments of the *UPS1* gene that were amplified by error-prone PCR. The plasmid encoding *URA3* was excluded from cells upon growth of the transformants in the presence of 5-fluorouracil-6-carboxylic acid (5′FOA). Plasmids encoding Ups1 mutants were isolated from growing cells and sequenced. **b** The frequency of mutations in Ups1 variants. Amino acids that were found mutated in >10 Ups1 variants are highlighted. Secondary structure elements in Ups1 are shown. **c** Growth of *ups2Δphb1Δ* cells expressing myc-tagged Ups2, Ups1 or the indicated Ups1 variants on selective medium containing glucose and 5′FOA. **d** Crystal structure of PA-bound Ups1[20] ([pdb:4ytx]) with amino acids highlighted that were found to be most frequently mutated

Ups1/PRELID1 subfamily (see Supplementary Figure 10C). E108 shows a similar degree of conservation, whereas other amino acids identified in our genetic screen are less well conserved. To examine the role of K58 in lipid specificity of the human PRELID1, we replaced this amino acid by valine or threonine and monitored the lipid transfer activity of purified mutant PRELID1–TRIAP1 complexes in vitro (Fig. 4a, b, Supplementary Figure 4A, B). While efficiently mediating the transfer of PA, PRELID1 showed a low PS transfer activity, similar to Ups1[14]. In contrast, substitution of K58 by valine or threonine

substantially increased the ability of PRELID1 to transfer PS, but only moderately affected its PA transfer activity (Fig. 4a, b, Supplementary Figure 4B). We therefore conclude that the role of K58 as a critical determinant of PS transfer activity is conserved from yeast to humans.

To substantiate these findings, we attempted to obtain ligand-bound crystal structures of human PRELI-like proteins. Although structures of phospholipid-free form of yeast Ups1 and human PRELID3a[19–21] have been determined, our initial electron density maps of the PRELID1 and PRELID3b structures showed that

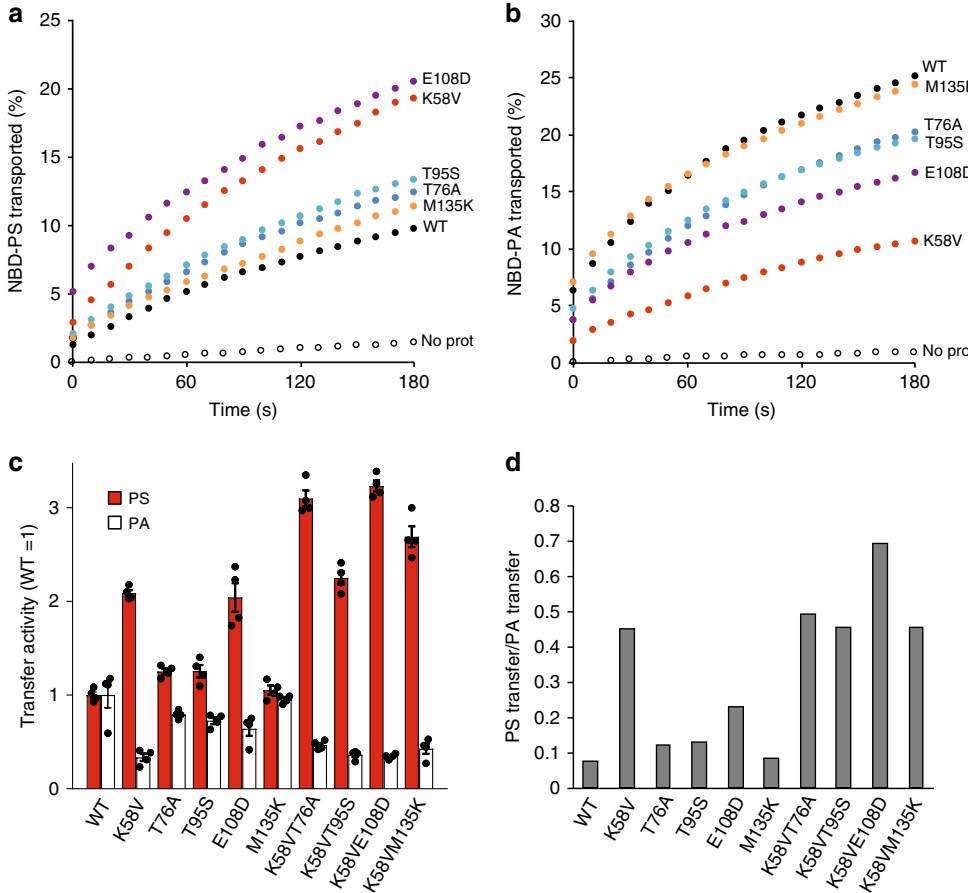

**Fig. 3** Lipid transfer specificity of Ups1 mutants. **a** PS transfer of wildtype (WT) and mutant Ups1–Mdm35 complexes in vitro. Recombinant protein complexes (50 nM) were incubated with donor liposomes (12.5 μM) containing NBD-PS and Rhodamine-PE (PC/PE/CL/Lac-PE/NBD-PS/Rhodamine-PE = 50/33/5/5/5/2 mol%) and acceptor liposomes (50 μM; PC/PE/CL/Lac-PE/PS = 50/35/5/5/5 mol%) at 20 °C and NBD fluorescence was monitored. **b** PA transfer of wildtype (WT) and mutant Ups1–Mdm35 complexes in vitro. Recombinant protein complex (12.5 nM) were incubated with donor liposomes containing NBD-PA and Rhodamine-PE (12.5 μM; PC/PE/CL/Lac-PE/NBD-PA/Rhodamine-PE = 50/33/5/5/5/2 mol%) and acceptor liposomes (50 μM; PC/PE/CL/Lac-PE/PA = 50/35/5/5/5 mol%) and NBD fluorescence was monitored. **c** Relative PS (red) and PA (white) transfer activates of Ups1 variants. Data were normalised to the activities of WT. Error bars represent mean ± SEM, $n = 4$. **d** PS transfer activities of Ups1 variants relative to their PA transfer activities. See also Supplementary Fig. 3

additional density existed within the cavity (Supplementary Figure 4C). Mass spectrometry (MS) identified that a fatty acid, likely an endogenous *E. coli* lipid molecule, was bound to PRELID1 and the abundant *E. coli* phospholipid phosphatidyl-glycerol (PG) was present in PRELID3b (Supplementary Figure 4D). We reasoned that the mutation of the critical K58 residue in PRELID1, which promoted PS transfer, would allow phospholipid exchange and subsequent crystallisation of a PS-bound ligand. We therefore incubated PRELID1–K58V with 1:2 DOPS-DDM micelles as described by Watanabe et al.[20] for Ups1 and PA, and following nickel purification, and SEC were able to confirm by MS that DOPS was able to displace the *E. coli* derived fatty acid, albeit incompletely (Supplementary Figure 4D). PRELID1–K58V incorporating DOPS formed readily crystals and diffracted to 2.98 Å. Although occupancy of the bound PS molecule was not expected at 100% due to partial exchange in our sample (Supplementary Figure 4D), a difference map corresponding to a larger ligand molecule appeared during initial refinement stages. A molecule of PS was modelled into the omit map observed, and this was further refined with ligand occupancy set to 0.3 in accordance with lipidomics analysis for the PS molecule (Fig. 4c). The PS molecule docked in the observed electron density was found enclosed within the cavity of PRELID1–K58V with the serine group positioned at the bottom of the pocket and

the acyl chain pointing towards the entrance of the pocket (Fig. 4c, d). The overall structure of the PRELI domain is largely unchanged in PRELID1–K58V mutant, showing a rmsd of 0.9 Å over the equivalent backbone atoms. The R25 side-chain, which is located near the PA phosphate of the Ups1-PA structure, is shifted towards the serine moiety in PRELID1–K58V (Fig. 4d). The smaller V58 side-chain of the mutant now allows the phosphodiester linkage to pass comfortably through and extend deeper into the cavity and the acyl chains of PS occupy the same channels as for PA. Interestingly, in the PA-bound Ups1 structure[20], additional space was observed beyond the phosphate deeper into the cavity and in our PS-bound PRELID1 mutant this is now occupied by the serine group from PS.

**Limiting PA transfer activity in Ups2/PRELID3b proteins**. Our yeast genetic screen combined with the structural analysis revealed conserved amino acids that limit the PS transfer activity of Ups1/PRELID1 proteins. However, the Ups1/PRELID1 variants retained significant PA transfer activity. The mutations thus broadened the substrate specificity of Ups1/PRELID1 proteins rather than converting them into PS-specific lipid transfer proteins. In contrast, the Ups2/PRELID3b proteins show high PS but low PA transfer activity[15,16] suggesting that additional mechanisms ensure their exclusive PS specificity.

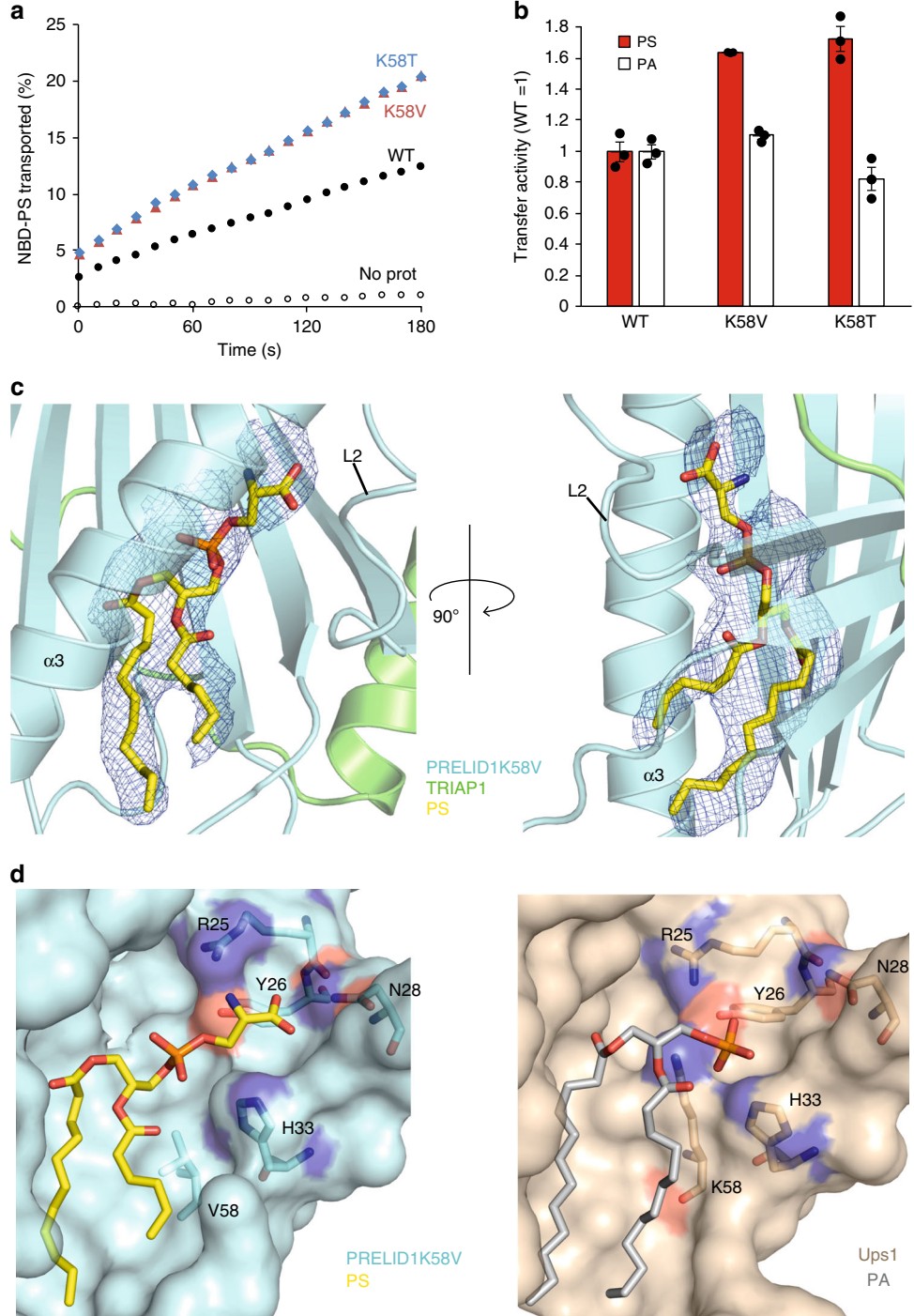

**Fig. 4** Characterisation of PS-bound PRELID3b mutants. **a** PS transfer by PRELID1, PRELID1–K58V (red triangles) or PRELID1-K58T (cyan diamonds). Assays are performed as in Fig. 3a, except that 120 nM protein was used. **b** PA and PS transfer activity of wildtype PRELID1 and mutant variants. Error bars represent mean ± SEM, $n = 3$. See also Supplementary Figure 4. **c** Left; PS headgroup orientation 1, with serine group protruding towards β-sheet of PRELID1–K58V (cyan). Right; PS headgroup orientation 2, with serine group protruding towards loop 2 of PRELID1. Electron density map for PS. The $F_o − F_c$ difference Fourier map was calculated by omitting PS, and is shown in cyan sticks surrounded by blue meshes countered at 3.0σ. **d** Comparison of PA-binding site in Ups1 and the PS-binding site of PRELID1-K58V-PS. Zoom in view of superposition of PRELID1–K58V-PS (cyan) and Ups1-PA (beige). All residues are numbered according to the protein sequence

We therefore searched for residues that limit PA transfer by Ups2 in a reverse genetic screen for Ups2 mutants that can substitute for Ups1 in vivo. Similar to our previous screen, we exploited the requirement of Ups1 for the growth of *phb1Δ* cells (Fig. 5a)[11]. *UPS2* was subjected to PCR mutagenesis and

expressed in *ups1Δphb1Δ* cells harbouring plasmid-borne *PHB1*. We screened for Ups2 variants that support cell growth upon loss of the *PHB1* encoding plasmid (Fig. 5a). We sequenced 112 mutant *UPS2* genes and identified in total 612 mutations (5.46 mutations per clone, Supplementary Data 1). The mutations

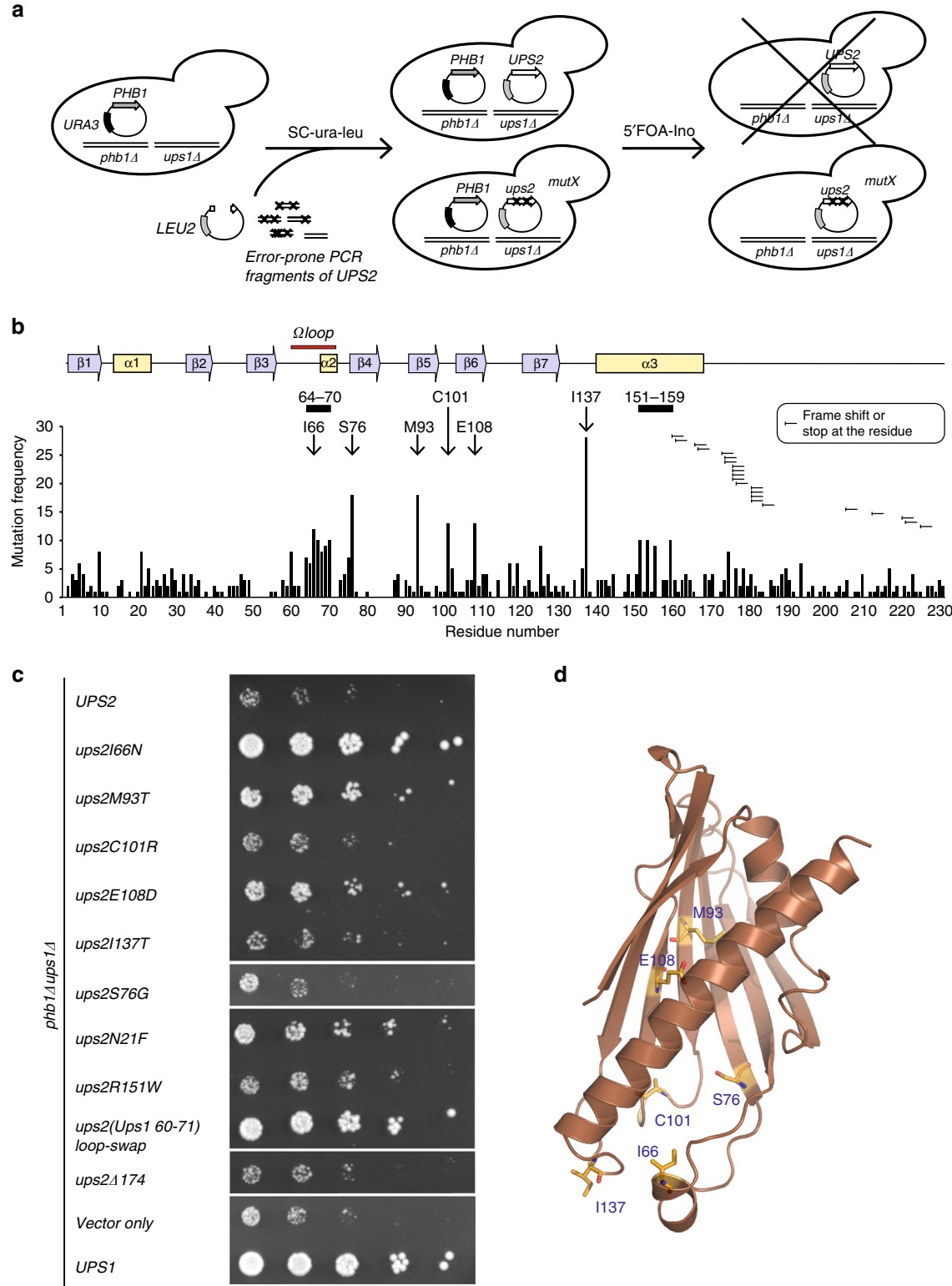

**Fig. 5** Identification of Ups2 variants transferring PA. **a** Reverse genetic screen for Ups2 variants substituting for Ups1. *ups1Δphb1Δ* cells harbouring plasmid-borne *PHB1* and *URA3* were transformed with a linearised yeast plasmid and with PCR fragments of the *UPS2* gene that were amplified by error-prone PCR. The plasmid encoding *URA3* was excluded from cells upon growth of the transformants in the presence of 5'FOA. Cells were grown in the absence of inositol to further suppress the growth of the host strain. Plasmids encoding Ups2 mutants were isolated from growing cells and sequenced. **b** The frequency of mutations in Ups2 variants. Amino acids that were found mutated in >10 Ups2 variants are highlighted. Secondary structure elements in Ups2 are shown. **c** Growth of *ups1Δphb1Δ* cells expressing myc-tagged Ups1, Ups2 or the indicated Ups2 variants on selective, inositol-free medium containing glucose and 5'FOA. **d** Structural model of Ups2 (based on the PRELID3b) with amino acids highlighted that were found to be most frequently mutated

affected a broad range of amino acid residues within Ups2, but those more frequently affected included I66, S76, M93, C101, E108 and I137 (Fig. 5b, Supplementary Table 2). Moreover, mutations accumulated in amino acid segments K64-L70 and R151-G159, which are part of the Ω loop and the C-terminal helix in Ups2, respectively (Fig. 5b, Supplementary Figure 5A). Finally, truncations of the poorly conserved and unstructured C-terminal region of Ups2 were frequently found (Fig. 5b, Supplementary Figure 5A).

To assess the importance of individual Ups2 mutations for the growth of ups1Δphb1Δ cells, we expressed Ups2 variants harbouring the most frequent mutations or lacking the C-terminal 53 amino acid residues of Ups2 in ups2Δphb1Δ or in ups1Δphb1Δ cells and monitored cell growth (Fig. 5c, Supplementary Figure 5B, 5C). With the exception of the Ups2 variant harbouring E108D, expression of all other Ups2 mutant forms allowed growth of ups2Δphb1Δ cells indicating that they retained PS transfer activity (Supplementary Figure 5B). In contrast to Ups2 or the C-terminal-truncated variant thereof, expression of most Ups2 variants also significantly improved growth of ups1Δphb1Δ cells (Fig. 5c), indicating that they can substitute for the loss of PA transfer by Ups1. We observed a moderately increased growth of ups1Δphb1Δ cells upon expression of Ups2 variants harbouring mutations in N21, M93 and E108 (Fig. 5c). Strikingly, Ups2 completely restored the growth of ups1Δphb1Δ cells, when I66 within the Ω loop was replaced with a less hydrophobic amino acid residue, suggesting a critical role of the Ω loop for substrate selection by Ups2. Consistently, many residues of the Ω loop of Ups2 were found to be mutated in our genetic screen for Ups2 variants that are able to substitute for Ups1 (Fig. 5b). We therefore replaced the complete Ω loop of Ups2 (amino acids 60–71) with the corresponding region of Ups1 (amino acids 60–71; see Fig. 6a) and examined the ability of the Ups2 loop-swap variant to restore growth of ups1Δphb1Δ cells. Cells harbouring the loop-swap variant grew similar to cells expressing Ups1 (Fig. 5c), indicating that the Ω loop of Ups1 confers PA transfer activity to Ups2.

Next, we monitored the PA/PS transfer activity of the loop-swap mutant of Ups2 in vitro. Ups2 showed an increased tendency to aggregate upon expression in a cell-free system or in E. coli, hampering the functional analysis of Ups2 variants. However, the solubility of Ups2 in vitro was significantly improved if the unstructured C-terminal region after W174 of Ups2 was removed (Supplementary Figure 6A), while preserving the functionality of Ups2 in vivo (Fig. 5c, Supplementary Figure 5B) and in vitro (Supplementary Figure 6B). We therefore introduced mutations in Ups2Δ174 and assessed the lipid transport activities of the variants in vitro. Ups2–Mdm35 complexes transferred PA only inefficiently when compared to PS (Fig. 6b, c, Supplementary Figure 6D). In contrast, we observed efficient PA transport by the loop-swap variant of Ups2 (~250% of Ups2), whereas its ability to transfer PS was significantly reduced (~30% of Ups2, Fig. 6b–d, Supplementary Figure 6D). Thus, replacement of the Ω loop of Ups2 by that of Ups1 converts the lipid transfer specificity of Ups2 from PS to PA (Fig. 6e, Supplementary Figure 6D). This indication is further substantiated by monitoring lipid transfer of non-fluorescent PA and PS (Supplementary Figure 6E, F). Accordingly, expression of the loop-swap variant of Ups2 significantly restored the CL levels in the cell lacking Ups1 (Fig. 6f), ensuring PA transfer activitiy of the variant in vivo. These experiments unravel the importance of the Ω loop for substrate specificity: while the Ω loop in Ups1 allows PA transfer, the corresponding region limits PA transfer activity in Ups2.

A sequence alignment of the Ω loops in Ups1/PRELID1 or Ups2/PRELID3b subfamilies did not reveal a sequence motif that

is specifically conserved within subfamilies (Fig. 6g). However, we noted that Ω loops in Ups2/PRELID3b are substantially more hydrophobic than those in Ups1/PRELID1 which contain (an) additional charged residue(s) in the region (Supplementary Table 3, Fig. 6g). Moreover, Ups2 variants that were identified in the genetic screen and able to substitute for Ups1 often showed reduced hydrophobicity in the Ω loop by introducing a positively charged residue (Fig. 6g). We therefore propose that the hydrophobicity and charge distribution of the Ω loop are critical determinants for the substrate specificity of Ups/PRELI lipid transfer proteins.

**Molecular simulations define the roles of the Ω loop.** To explore the mode of interaction between of PRELID3b–TRIAP1 and PRELID1–TRIAP1 complexes with a phospholipid bilayer, we performed coarse-grained molecular simulations with PS- and PA-containing membranes (Supplementary Figure 7, Supplementary Table 4, Supplementary Movies 1 and 2). The complexes were placed ~10 nm away from the bilayer and allowed to freely diffuse in solution. The simulations were performed with a single lipid-bound in the cavity (POPS for PRELID3b and POPA for PRELID1) (Supplementary Table 4, Supplementary Figure 7A, 7B). In all cases, the lipids remained bound in the cavity whilst the complex was free in solution (Fig. 7c, d, Supplementary Figure 7C, 7D).

The complexes contact the membrane several times before converging upon the stable binding mode, in which large aromatic residues are buried in the membrane (Fig. 7a, b, Supplementary Figure 7). The initial attachment to the membrane involves burial of one or more of these 'anchoring' residues before the insertion of the Ω loop and part of the C-terminal α3 helix in the bilayer (Fig. 7b, Supplementary Figure 8A). The next stage of the binding series involves an opening motion of the Ω loop, which allows the bound lipid to move down from the primary binding pocket, into a secondary site, before detaching fully. The pathway of the lipid during this process may be mapped in terms of close contact points between the lipid headgroup and the PRELI domain (Fig. 7c, d). These plots highlight several residues that were identified in the genetic screens as forming key contacts with the lipid substrate within the primary site. Residues in contact with the lipid during the exit pathway may also have important roles in substrate uptake and selectivity. K58 was notable in the PA release pathway as it maintained contact with the PA headgroup in the secondary site.

The binding orientation of PRELID1 and PRELID3b on the membrane is such that the lipid is primed in the preferred orientation to be readily released from the binding channel into the membrane (Fig. 7a, Supplementary Figure 7 and 8A, see Supplementary Movies 1 and 2). The hydrophobic lipid tails are able to partition into the bilayer, while the headgroup is released in the final step from the complex at the approximate height of the rest of the membrane lipid headgroups.

Interestingly, we found that PA was released from the complex with only partial burial of the protein, limited to the Ω loop and one end of the adjacent C-terminal helix, which are deeply buried into the hydrophobic region of the membrane (Fig. 7b, Supplementary Figure 8A). PS release, on the other hand, involved a more extensive overall interface of the protein surface on the membrane at the point of lipid release, consistent with the increased hydrophobicity of the Ω loop in the PS-specific Ups2/PRELID3b proteins. This could be quantified by measuring the number of contacts between each residue and the hydrophobic region of the membrane (Supplementary Figure 8A).

To further characterise this difference in behaviour, we performed simulations of the PRELID1–K58V complex with

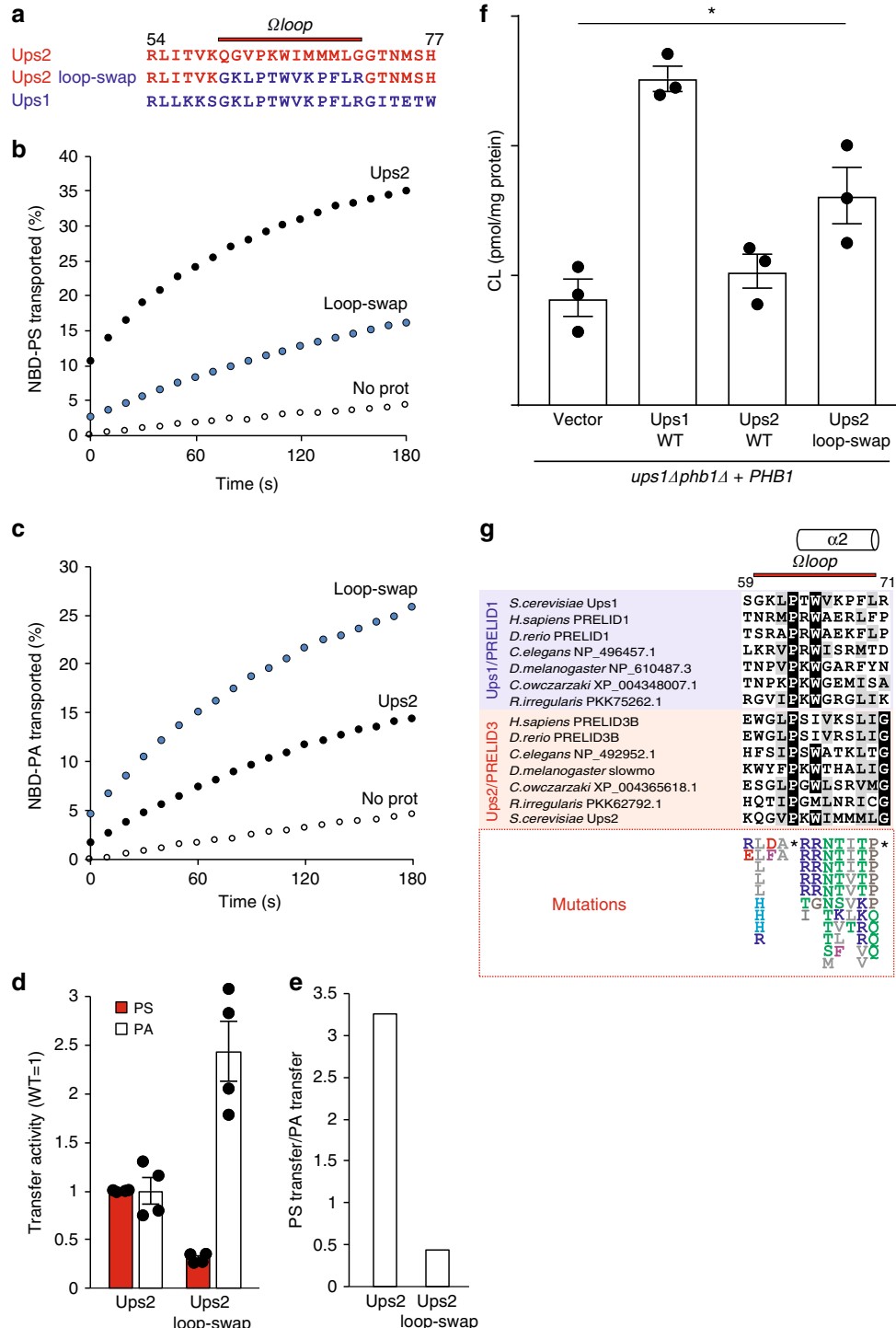

**Fig. 6** Lipid transfer specificity of Ω loop mutants of Ups1 and Ups2. **a** Amino acid sequence of the region surrounding the Ω loop in Ups1 and Ups2 and of the Ups2 loop-swap mutant. **b** PS transfer by Ups2 and loop-swap variant. The Ω loop was exchanged in a truncated version of Ups2 (Δ174) that is functionally active but shows reduced tendency to aggregate in vitro (Supplementary Figure 6A). Recombinant protein complexes (50 nM) were incubated with donor liposomes (12.5 μM) containing NBD-PS and Rhodamine-PE (PC/PE/CL/Lac-PE/NBD-PS/Rhodamine-PE = 50/23/15/5/5/2 mol%) and acceptor liposomes (50 μM; PC/PE/CL/Lac-PE/PS = 50/25/15/5/5 mol%) at 25 °C and NBD fluorescence was monitored. **c** PA transfer by Ups2 and the Ups2 loop-swap variant. Recombinant protein complexes (50 nM) were incubated with donor liposomes (12.5 μM) containing NBD-PA and Rhodamine-PE (PC/PE/CL/Lac-PE/NBD-PA/Rhodamine-PE = 50/23/15/5/5/2 mol%) and acceptor liposomes (50 μM; PC/PE/CL/Lac-PE/PA = 50/25/15/5/5 mol%) at 25 °C and NBD fluorescence was monitored. **d** PS and PA transfer activities of the Ups2 loop-swap variant normalised to control. Error bars represent mean ± SEM, n = 4. **e** PS/PA transfer activity ratio of Ups2 and the loop-swap variant. **f** Total CL levels in ups1Δphb1Δ cells expressing Phb1 and Ups1, Ups2 or Ups2 loop-swap. N = 3. Error bars represent mean ± SEM. *p < 0.05. **g** Sequence alignment of Ω loop regions of members of the Ups1/PRELID1 and Ups2/PRELID3b subfamilies in the indicated eukaryotic organisms. Mutations in Ω loop of Ups2 that were identified in the genetic screen are indicated below. Asterisks refer to the residues that were not found mutated

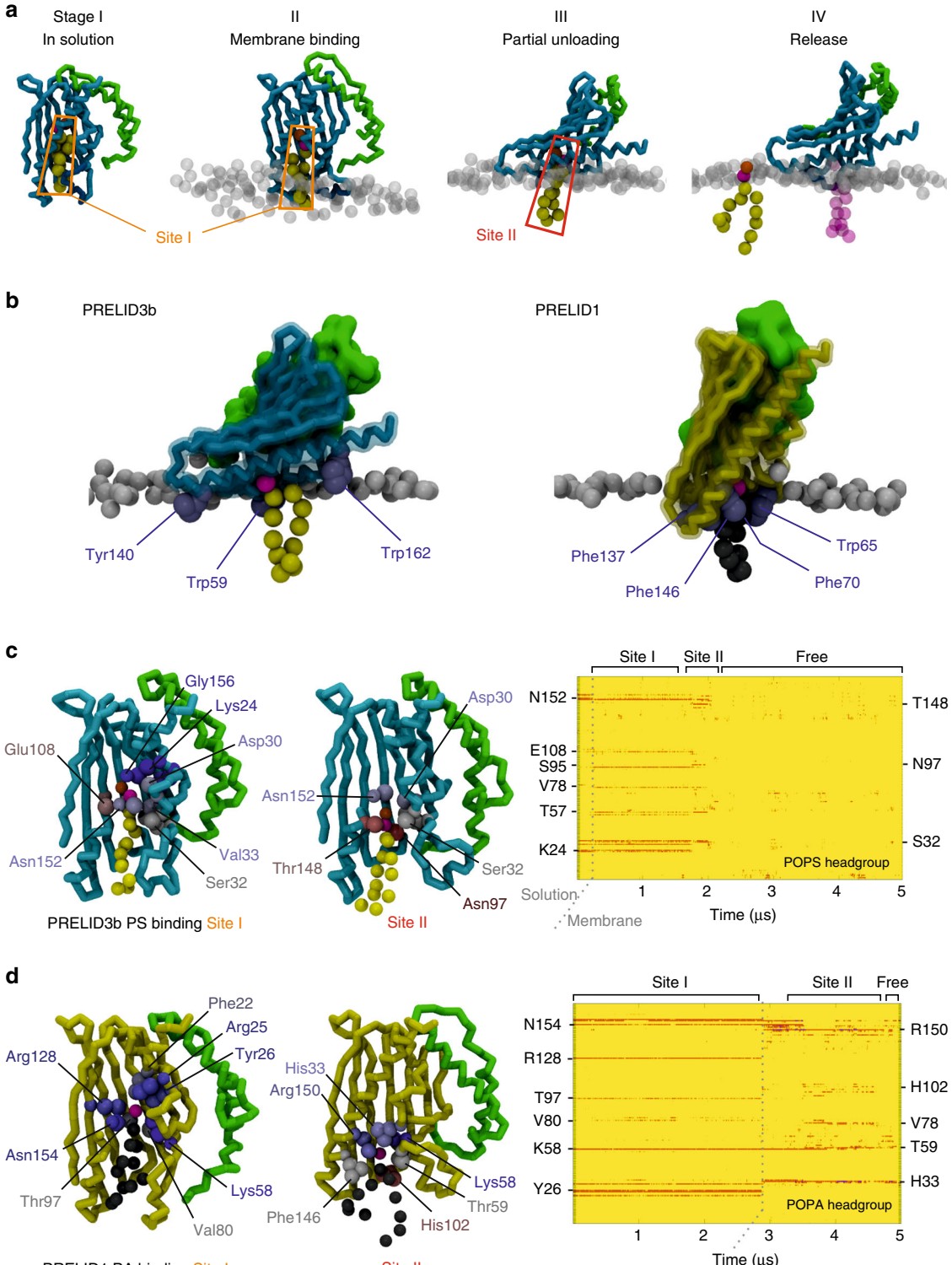

**Fig. 7** Coarse-grained simulations of membrane association. **a** Stages of membrane binding and PS release by the PRELID3b–TRIAP1 complex. Bound PS molecule coloured yellow except for phosphate (magenta) and serine (orange) particles. Final step shows a new PE lipid (magenta, transparent) bound in Site II. PRELID3b (cyan) and TRIAP (green) are shown as a backbone trace, with the Ω loop shown in dark blue. Membrane lipid phosphate groups are shown as grey spheres. **b** Orientation of PRELID3b (cyan) and PRELID1 (yellow) on the membrane whist lipid is in Site II. Anchoring aromatic residues are shown as purple spheres. **c** Key residues of PRELID3b in contact with PS phosphate group are labelled for each binding site. Heatmap on the right shows contacts over time for the first simulation. Residues are coloured according to proportion of time in contact with lipid phosphate group over simulation (red minimum, blue maximum). **d** As **c** but for PRELID1 and PA. PA is coloured grey with magenta phosphate group

POPS (Supplementary Table 4). PRELID1–K58V was found to maintain the same upright orientation on the membrane indicating that the flattened orientation observed for PRELID3b during PS release is a property of the protein rather than a necessary requirement of the lipid substrate (Supplementary Figure 8B). Furthermore, key contacts formed by PRELID1 with the PA headgroup were lost for PRELID1–K58V with PS, which were not compensated for by the serine moiety (Supplementary Figure 8B). We also tested the PRELID3b–PRELID1-loop-swap mutant with bound PA (Supplementary Figure 9, Supplementary Table 4). This exhibited PA release in either an upright or flat conformation, which highlights the influence of the Ω loop in membrane-binding orientation. As a final comparison, we also simulated the behaviour of the Ups1–Mdm35 PA-bound complex and found the same upright orientation on the membrane (Supplementary Figure 9). Following lipid release, the open secondary binding site could either be occupied by new lipids or the Ω loop closes again. The final snapshots of each simulation are shown in Supplementary Figure 9, highlighting the stochastic nature of the process.

## Discussion

The Ups/PRELI lipid transfer proteins are widely conserved in eukaryotes and can be classified into two subfamilies by sequence similarity (Ups1/PRELID1 and Ups2/PRELID3; see Supplementary Figure 10C)[26]. The crystal structures of PRELID1[1–173] and PRELID3b, together with previously reported PRELID3a, means that structural information is available for all human homologues of this lipid transfer protein family, which provides the framework to deepen our understanding of the structural determinants for lipid specificity. A structural comparison of the complete Ups/PRELI-like family highlights a conformationally flexible entrance to the phospholipid binding cavity. The crystal structures display a range of conformations for the Ω loop and the adjacent C-terminal helix α3 which cap the cavity (Fig. 1c, Supplementary Figure 10A). These two features are observed to progressively shift apart, akin to an index finger and thumb in a pinching motion, suggesting that conformational changes within the Ω loop and the C-terminal helix are important role for efficient substrate capture and delivery. The interface with TRIAP1 or Mdm35 remains unaltered despite these conformational differences within the PRELI domain. PRELID1[1–173] exhibits the most closed of all the structures while the structure of PRELID3b displays the most open entrance to the lipid binding cavity, and this likely contributes to its ability to readily accept the larger headgroup of PS.

The strong conservation throughout the PRELI family hampered the identification of critical residues involved in substrate selection from structural data alone. Insight came from reverse genetic screens in yeast for Ups1 and Ups2 variants with broadened or swapped lipid specificity. The gain-of-function Ups1 mutants from this approach can be categorised into two different functional regions: (a) residues facing to the inner substrate-binding cavity that tailor the cavity for selective accommodation of PA (K58, T95, E108) and (b) residues proximal to the Ω loop (K58, T76) and C-terminal α3 helix that likely have a role together in lipid extraction and capping an occupied binding cavity (F133, M135).

The critical K58 residue is conserved within Ups1/PRELID1 subfamily proteins but not in Ups2/PRELID3b, which harbour small polar or hydrophobic residues at this position. This is consistent with the role of the lysine residue being the key determinant for PA-specific transfer. K58 points into the lipid binding cavity of the PRELI domain and together with R25 make a specific electrostatic interaction with the negatively charged

phosphomonoester of PA (Supplementary Figure 10B). Hydrogen bond interactions from basic side chains stabilise the −2 charge state of PA and this reinforces the electrostatic interaction and provides specificity for PA. In PS, the phosphate forms a phosphodiester linkage with a terminal serine moiety and therefore has a maximal charge state of −1 with a distinct charge density. Accommodation of this larger headgroup would not only be hindered by the bulkiness of the lysine side-chain, as observed in our structural observations (Fig. 4c, d), but the positive charge of K58 may capture the single-charged PS headgroup inappropriately and prevent it from being fully coordinated deeper within the cavity. Exchange of this residue for a small polar or hydrophobic side-chain would remove these limiting factors and promote accommodation of the PS headgroup. The side chains of T95 and E108 also protrude into the substrate-binding cavity near the PA headgroup and thus the changes to smaller residues at these positions (T95A or E108D) would also ease accommodation of the larger PS. The hydrophobic residues F133 and M135 are located at the beginning of the C-terminal α3. The role of these residues could be to either promote the interaction of the PRELI domain with the membrane prior to lipid loading/release or stabilise a key intermediate that allows access of the lipid molecule to the PRELI cavity. Consistent with this assumption, corresponding amino acid residues in PRELID1 are buried most deeply into the lipid bilayer while PA release in MD simulations. T76 flanks the Ω loop and makes direct contacts with one of the acyl chains of PA and therefore its role in substrate selection likely reflects an altered positioning of the acyl chains in a PS-bound PRELI domain. Molecular dynamics simulations with phospholipid-loaded PRELIDs provide further evidence for this, as a significant conformational change is observed within the Ω loop upon engagement with the bilayer, bringing T76 into direct contact with the PA headgroup during the final step of phospholipid release.

Our genetic screen for Ups2 mutants that are able to substitute for Ups1 revealed the primary importance of loop regions (the Ω loop, β5-β6, β7-C3) and the C-terminal helix in PS-specific transfer. The Ups2 variant carrying Ω loop from Ups1 is able to transfer PA, highlighting a pivotal role for the Ω loop in substrate selection. Notably, K58 is located in close proximity to the Ω loop raising the possibility that conformational changes in the loop region are communicated to residues involved in headgroup recognition of the lipid. Molecular simulations confirm that K58 maintains contact with the lipid headgroup throughout, from the bound conformation identified in the crystal structure via its route through the entire exit pathway. The Ω loop and the short alpha helix (α2) in Ups/PRELI proteins contain several hydrophobic residues that ensure lipid binding. However, the overall hydrophobicity in this region is significantly higher in Ups2/PRELID3b than in Ups1/PRELID1, which harbours additional charged residues in this region (Fig. 6g and Supplementary Table 3). In MD simulations, a more extensive interface with the membrane is generated with PRELID3b over PRELID1, reflecting the higher hydrophobicity (Fig. 7b, Supplementary Figure 8A). Consistent with a prominent role for substrate selection, the majority of mutations found in the Ups2 screen reduce the hydrophobicity of this region. Moreover, the comparison of the PRELI structures shows significant conformational plasticity in this region and suggests that a large conformational change is important for specific lipid transfer. We propose that these regions, especially the Ω loop, are involved in extraction of lipid molecules from membrane and loading of the lipid into inner cavity for specific capture. The small headgroup of PA give a cone-shaped structure, which prevents tight packing within a bilayer and presents a lower barrier to membrane insertion by PA-binding proteins. In the case of PS, which is more cylindrical,

headgroup packing is more efficient, and the bilayer leaflet less readily penetrated. It is therefore conceivable that higher hydrophobicity of the Ω loop and neighbouring regions in PS-specific PRELI domains is required to facilitate the necessary penetration into the acyl region of PS-enriched membranes and the correct positioning of the headgroup in the binding cavity. This is consistent with the previous observations that mutations within the Ω loop retard extraction of PA from membrane by Ups1–Mdm35[19,20]. Furthermore, the increased hydrophobicity of the PS-specific Ω loop region likely inhibits release of the PRELI domain from a PA-enriched membrane, thereby retarding PA transport. Indeed, the Ups1 variant carrying the hydrophobic Ω loop of Ups2 was not able to transfer PA efficiently (Supplementary Figure 6G, 6H). Moreover, the Ups1 loop-swap variant did not transfer PS as well, indicating that efficient PS transfer requires both the hydrophobic Ω loop and the substrate-binding cavity that is tailored for PS.

Coarse-grain simulations have allowed us to explore molecular details of lipid unloading by PRELI domains beginning from free complexes in solution, requiring long timescales that would not be feasible for atomic simulations. However, we respect their limitations, particularly the need for maintaining secondary structure, changes in which may have a role in the membrane association as found for PITPα[27]. Future work could exploit atomic resolution simulations using enhanced sampling methods in order to further characterise the dynamic Ω region in greater detail, and be supplemented with experimental restraints, such as those available from NMR. Using CGMD we were able to observe lipids entering site II, however no lipids were fully extracted from the membrane on the 5 μs timescale. However, assuming the loading and unloading pathways of the lipid are similar, we expect that the partial desorption of a lipid to site II may drastically lower the energy barrier for the next loading step into the lipid binding cavity, and the tailored environment in the cavity may further support loading of a specific lipid molecule.

Together, our results define the important structural determinants for specific phospholipid transport within mitochondria. In addition to headgroup recognition, we highlight a role for conformational plasticity and hydrophobicity of the Ω loop gate in the PRELI domain, in regulating membrane targeting and capture of specific lipids. Notably, a similar role of flexible loop regions has been postulated for other lipid transfer proteins of the STARkin superfamily, pointing to similar structural principles of substrate selection[27–32].

## Methods

**E. coli strains and growth conditions**. E. coli strains used for expression of recombinant proteins are Shuffle T7 (NEB) and Rosetta-gami2 (DE3) (Merck). For preparation of DNA or cloning of genes XL1 Blue strain was used. In general, these E. coli strains were cultivated in LB medium containing antibiotics (100 μg/ml ampicillin) for selection of plasmid at 37 °C otherwise indicated in method details.

**S. cerevisiae strains and growth conditions**. Yeast strains used are based on S288c and listed in Supplementary Table 5. Strain PY51 were generated by tetrad dissection of a haploid carrying *HYG*, *NAT* and *URA3* after crossing PD49 (*ups1Δ*) and CG409 (*phb1Δ* + pCM189-PHB1). Cells harbouring *UPS1* or *UPS2* and *PHB1* were cultivated in synthetic complete medium lacking uracil and leucine supplemented with 2% glucose as a carbon source. For the experiments checking the growth phenotype of cells with impaired Ups1 function, *myo*-inositol was omitted from medium. See method details for precise growth conditions used in the genetic screens.

**Protein expression and purification for structural studies**. A C-terminal truncation of PRELID1 spanning residues 1–173 (PRELID1[1–173]) with an N-terminal histidine tag was cloned into the pET-duet-1 vector, and Cys112 and Cys115 were substituted for Ser by Q5 site-directed mutagenesis (NEB). TRIAP1 was cloned into the higher copy vector pRSF1b with an N-terminal histidine tag. The constructs were co-transformed into SHuffle E. coli cells, and co-expressed after induction overnight at 16 °C with 0.1 mM IPTG at a cell density of OD₆₀₀ 0.6, in 1 l LB pre-

inoculated cultures. Cell pellets were resuspended in 20 ml of lysis buffer (50 mM Tris-HCl pH 8.0, 200 mM NaCl) supplemented with 1× EDTA-free protease inhibitor cocktail (Sigma) and then disrupted by sonication. Lysates were centrifuged at 15,000 rpm for 30 min at 4 C and the clarified lysate was loaded on a Ni-NTA agarose column pre-equilibrated with 20 column volumes (CV) of wash buffer (50 mM Tris- HCl pH 8.0, 200 mM NaCl, 20 mM imidazole). This was then washed with 20 CV of wash buffer and eluted with 5 CV of elution buffer (50 mM Tris-HCl pH 8.0, 200 mM NaCl, 300 mM imidazole). Following nickel affinity purification the sample was further purified using a Superdex 75 Hi Load preparative gel filtration column (GE Healthcare) in gel filaration buffer (50 mM Tris, pH 8, 200 mM NaCl), to separate PRELID1[1–173]/TRIAP1 complexes from the excess free TRIAP1 in the sample. The PRELID1[1–173]/TRIAP1 complexes were concentrated to 26 mg/ml using an Amicon 30 kDa MWCO ultracentrifugation unit (Millipore). For crystallisation, the concentrated protein sample was mixed 1:1 with crystallisation liqour (100 mM CHES/ NaOH pH 9.5, 200 mM NaCl, 40% (v/v) PEG 300) and dispensed into 1 μl sitting drops in the wells of MRC-MAXI 24 well plates, alongside a reservoir of 200 μl crystallisation liquor. Plates were left to equilibrate for 9 days at 20 °C. crystals were harvested and flash frozen in liquid nitrogen for data collection.

For expression of PRELID3b–TRIAP1 complex, TRIAP1 was cloned into vector pMAL(X)E including an N-terminal Maltose binding protein, and PRELID3b was cloned into vector pRSF-2. These were co-transfrmed in SHuffle E. coli cells, then co-expressed, harvested and Ni-NTA purified as described for PRELID1–TRIAP1. PRELID3b–MbpTRIAP1 complex was pulled down via N-terminally his-tagged PRELID3b. Following nickel affinity purification the sample was gel filtered using a Superdex 200 Hi Load preparative column (GE Healthcare). Fractions of the first peak corresponding to dimer of heterodimeric PRELID3b–MbpTRIAP1 complex were pooled together and exchanged into crystallisation buffer (20 mM Tris-HCl pH 8.0, 50 mM NaCl, 5 mM Maltose). PRELID3b–MbpTRIAP1 complexes were concentrated to 26 mgml⁻¹ using an Amicon 30 kDa MWCO ultracentrifugation unit (Millipore). For crystallisation, the concentrated protein sample was mixed 1:1 with crystallisation liqour (Tacsimate (60% v/v) pH 7) and dispensed into 1 μl sitting drops in the wells of MRC-MAXI 24 well plates, alongside a reservoir of 200 μl crystallisation liquor. Plates were left to equilibrate for 4 days at 20 °C. Crystals were harvested and flash frozen with 25% glycerol in liquid nitrogen for data collection.

The PRELID1[1–173]–TRIAP1 K58V mutant was cloned from the wildtype using a Q5 Site-Direction Mutagenesis Kit (NEB). The complex was expressed and purified as described for the wildtype. Following nickel purification the eluted sample was incubated with a 10-fold excess of DOPS solubilized in dodecyl maltoside (Anatrace). The sample was then gel filtered into 20 mM Tris-HCl pH 8.0, 200 mM NaCl buffer and concentrated to 25 mg/ml. Crystallisation screens were performed as described for PRELID1. Crystals formed after 5 days in 200 mM calcium acetate hydrate, 100 mM sodium cacodylate pH 6.5 and 40% v/v PEG 300 (Molecular Dimensions). Crystals were harvested and flash frozen in liquid nitrogen.

**Data collection and processing**. Data were collected for PRELID1[1–173]–TRIAP1 and PRELID1[1–173]–TRIAP1 K58V crystals on the I03 beamline (Diamond Light Source, UK) and the best dataset in each instance was selected for data reduction. Data from 1000 images were integrated and scaled using xia[33], in the space group P 6₃ 2 2.

A single PRELID3b–TRIAP1 crystal diffracted to 2.9 Å resolution on the I04 beamline (Diamond Light Source, UK) with wavelength of 0.980 at 100 K with an oscillation angle of 0.2° per frame, using a Pilatus 6M-F. A total of 1000 images were collected and merged. The dataset was auto processed using xia2 in space group I 2 2 2.

**Structure determination**. The Program Phaser MR[34] was used to solve the structure using the coordinates of previously solved PRELID3A–TRIAP1 structure[19]. For the PRELID1[1–173]–TRIAP1 structure, a single-copy poly-alanine model of PRELID3A–TRIAP1 was used as a search model for molecular replacement followed by 5 cycles of REFMAC5[35] to produce a $2F_o − F_c$ and $F_o − F_c$ electron density map for model building. For the PRELID3b–TRIAP1 structure, MBP, TRIAP1 and PRELID3A were introduced as three individual search ensembles for molecular replacement. At the end of the process a map coefficient for all three imported components was calculated into P I 2 2 2 space group. The PRELID1[1–173]–TRIAP1 was used as the search model for PRELID1[1–173]K58V–TRIAP1.

**Model building and refinement**. Initial models were built using Buccaneer[36], then modified in COOT[37] with iterative cycles of model building and refinement for the PRELID1[1–173]-TRIAP1 structure. For PRELID3b–TRIAP1, the initial PRELID3A–MbpTRIAP1 model fitted well into the electron density, apart from the long C-terminal α3 helix, which presented a different conformation and revealed an intermolecular association occurring between electron density of the PRELID3b model and a symmetry related PRELID3b molecule via swapping their C-terminal α3 helices. As a consequence, helix α3 was manually built in COOT fitting a total of

34 amino acids. Model building and refinement was carried out using REFMAC5[35] and also Phenix[38].

**Model analysis and validation**. The structures were validated using COOT and MolProbity[39]. The atoms in the backbone and side chains fit in the electron density and orientations of side-chain were adjusted by rotamer analysis. The Ramachandran analysis is reported in Supplementary Table 6.

**Cloning of genes in yeast expression vectors**. C-terminally *myc*-tagged variants of yeast *UPS2* had been cloned in YCplac111 vector under *ADH1* promoter[11]. c-*myc*-tagged variants of yeast *UPS1* was amplified by PCR using primers TL4606 and TL4607 (primer sequences are listed in Supplementary Table 7) and cloned into the *Sma*I and *Hin*dIII sites in YCplac111 under *ADH1* promoter. Point mutants were generated by site-directed mutagenesis PCR using Pfu Turbo DNA polymerase (Agilent) and the primers listed in Supplementary Table 7.

**Reverse genetic screens in yeast**. To isolate Ups1 mutants suppressing the severe growth phenotype of the cell lacking *PHB1* and *UPS2*, a PCR fragment encoding the whole *UPS1* gene was amplified from the plasmid YCplac111P$_{ADH1}$-*UPS1*myc by error-prone PCR (in the presence of 0.25 mM MnCl$_2$ and 0.05 U/µl Taq polymerase) with the primers TL 11721 and TL 11722. GC410 (*ups2Δphb1Δ* pCM189-*PHB1*) cells grown in SCD containing 1 µg/ml doxycycline were isolated and transformed with 1 µg of the PCR fragment and 200 ng of a 7.5 kb fragment resulting from restriction digest of YCplac111P$_{ADH1}$ by *Sma*I and *Hin*dIII. Transformation mix was spread on ten SCD-leucine plates containing 60 µg/ml uracil and 1 µg/ml doxycycline. After incubation for 24 h at 30 °C, transformants were replica plated on SCD-leucine plus 60 µg/ml uracil, 1 µg/ml doxycycline and 0.1% 5′-Fluoro Orotic Acid (5′-FOA). The replica plates were incubated for 48 to 72 h at 30 °C. Appeared colonies on the plates (in total 133 colonies) were picked and streaked on SCD-leucine plus 60 µg/ml uracil, 1 µg/ml doxycycline and 0.1% 5′-FOA plates to purify as well as on SCD-uracil to check the loss of *PHB1* plasmid. Colonies appeared on the plate were collected, and subjected to plasmid extraction using E.Z.N.A plasmid mini kit (OMEGA bio-tek). To amplify the recovered plasmids, they were individually transformed in *E. coli* XL1blue and the transformants were subjected to plasmid extraction by the kit. *UPS1* genes on the plasmids were sequenced to identify mutations (in total 119 clones).

To isolate Ups2 mutants suppressing the severe growth phenotype of the cell lacking *PHB1* and *UPS1*, a PCR fragment encoding the whole *UPS2* gene was amplified from the plasmid YCplac111P$_{ADH1}$-*UPS2*myc by error-prone PCR (in the presence of 0.25 mM MnCl$_2$ and 0.05 U/µl Taq polymerase) with the primers TL 12090 and TL 12091. PY51 (*ups1Δphb1Δ* pCM189-*PHB1*) cells grown in SCD containing 1 µg/ml doxycycline were isolated and transformed with 1 µg of the PCR fragment and 200 ng of a 7.5 kb fragment resulting from restriction digest of YCplac111P$_{ADH1}$ by *Sma*I and *Hin*dIII. Transformation mix was spread on 10× SCD-leucine plates containing 60 µg/ml uracil, 7.5 µM *myo*-inositol (1/10 of standard concentration of myo-inositol in SC medium) and 1 µg/ml doxycycline. After incubation for 24 h at 30 °C, transformants were replica plated on SCD-leucine plus 60 µg/ml uracil, 1 µg/ml doxycycline and 0.1% 5′-Fluoro Orotic Acid (5′-FOA) but lacking inositol. The replica plates were incubated for 48 to 72 h at 30 °C. Appeared colonies on the plates were picked (in total 112 colonies) and streaked on SCD minus leucine and inositol plus 60 µg/ml uracil, 1 µg/ml doxycycline and 0.1% 5′-FOA plates to purify as well as on SCD-uracil to check the loss of *PHB1* plasmid. Plasmids were recovered from growing colonies. *UPS2* genes on the plasmids were sequenced to identify mutations (in total 106 clones).

**Cloning of genes for biochemical studies**. N-terminally His-tagged variants of yeast *UPS1* or *UPS2* (ΔCys; all four cysteine codons have been replaced by serine codons) were cloned in pETDuet-1 vector (Merck) together with Mdm35[13,15]. N-His-tagged variants of human PRELID1 or PRELID3b were cloned in pETDuet-1 vector together with TRIAP1[14,15]. Point mutations were introduced by site-directed mutagenesis using *Pfu Turbo* DNA polymerase (Agilent) using the primers listed in Supplementary Table 7. For expression of the loop-swap variant of yeast Ups2 together with Mdm35, codon-optimised *MDM35* gene was synthesised by GeneArt gene synthesis service (Thermo Fisher Scientific) and cloned into pETDuet-1 vector between *Nde*I and *Xho*I sites. Codon-optimised His*UPS2*, His*UPS2* variant carrying Ω loop region of Ups1 (amino acids 60–71), His*UPS1* and His*UPS1* variants carrying Ω loop region of Ups2 (amino acids 58–71 or 60–71) were synthesised by GeneArt gene synthesis service and cloned in pETDuet-1-*MDM35* between *Nco*I and *Hin*dIII sites.

**Synthesis and purification of Ups1–Mdm35**. N-terminally his-tagged Ups1 together with Mdm35 were expressed in a bacterial lysate-based, continuous-exchange cell-free expression system as described in detail in Schwarz et al.[40]. using pETDUET-his*UPS1-MDM35* or a variant thereof encoding *ups1* mutants. Protein expression was achieved by shaking at 30 °C for 16 h in a protein expression mix based on S30 lysate of *E. coli* with T7 polymerase (Cube Biotech, 35% (v/v) in reaction mix) in the presence of 15 µg/ml plasmid DNA but devoid of DTT. Protein purification of His-Ups1–Mdm35 complex was performed as described[15]. The purified protein sample was subjected to dialysis in storage buffer (10 mM Tris-HCl pH 7.4, 100 mM NaCl, 1 mM EDTA) for 4 h. Protein concentration was determined by absorbance at 280 nm and extinction coefficient of the protein complex.

**Expression in *E. coli* and purification of Ups2–Mdm35**. Ups2–Mdm35 and its variants were expressed in *E. coli* Rosetta-gami2 (DE3)(Promega). After incubation at 37 °C for 3.5 h (OD600 ~0.5), the culture was shifted to 18 °C for 1 h and then Ups2 and Mdm35 were expressed by adding IPTG (0.2 mM) for 14 h. *E. coli* cells were lysed in buffer C [50 mM Tris/HCl, pH 8, 500 mM NaCl, 1× Halt protease inhibitor mix, 1 mM PMSF, 20 mM imidazole, 100 U/ml Dnase I, 1 mM MgCl$_2$, 25 ml/g wet cell weight] by Emusiflex C-5 (Avestin). The lysate was spun at 30,000 × *g* for 20 min and the supernatant was recovered to a tube. Ni-sepharose high performance beads (GE healthcare) were added to the lysate (20 µl/ml) and his-tagged proteins were captured to beads by incubation for 1 h at 4 °C. After incubation, beads were recovered by centrifugation (1000×*g*, 2 min), washed four times with 50× beads volume of wash buffer C (20 mM Tris-HCl pH 8.0, 500 mM NaCl, 30 mM imidazole) and then proteins were eluted in elution buffer C (20 mM Tris-HCl pH 8.0, 500 mM NaCl, 300 mM imidazole, 20% glycerol). The elution was subjected to size exclusion chromatography using Superdex 200 increase 10/300 column (GE healthcare) in buffer B (10 mM Tris-HCl pH 8.0, 500 mM NaCl) immediately. NBD-PS transfer activities were determined in elution fractions and fractions containing highest PS transfer activity were combined. Protein concentrations in samples were determined by absorbance at 280 nm and extinction coefficient of the protein complex. Samples were diluted to 10 µM in buffer B and divided into small aliquots and then stored at −80 °C.

**Lipid transfer assays**. Lipid transfer of NBD-PA (18:1–12:0 NBD-PA, Avanti polar lipid 810176) or NBD-PS (18:1–12:0 NBD-PS, Avanti plar lipid 810195) by Ups1-Mdm35, Ups2–Mdm35 or PRELID1–TRIAP1 were tested in 120 µl assay buffer (20 mM Tris-HCl, pH 7.4, 100 mM NaCl, 1 mM EDTA) at 20 °C in the presence of 12.5 µM donor liposomes and 50 µM acceptor liposomes (total lipid concentration, 0.1 µm diameter, composed of synthetic phospholipids containing oleic acid in its acyl chains if not indicated). The compositions of liposomes and the concentration of proteins used in individual experiment were indicated in figure legends. Dequenching of NDB fluorescence upon addition of accepter liposomes was monitored in SpectraMax Paradigm microplate reader (Molecular Devices). Because of the initial delay (ca. 10 s) until recording the first data point after starting the transfer reaction, the NBD fluorescence at time point zero varies depending on the transfer activities in the reaction. NBD fluorescence of the liposomes lacking the quencher Rhodamine-PE was set to 100% and molecules of NBD-PA/PS transported per complex per second were calculated from the increase of NBD fluorescence in the initial 30 s.

Lipid transfer of non-fluorescent, endogenous PS or PA by Ups1-Mdm35, Ups2–Mdm35 or their variants were assessed as described above, except having non-fluorescent PA or PS and tracer lipids (17:0 PC and 17:0 PE) in the liposomes and the increase in reaction size (400 µl). Precise conditions used in individual experiment were indicated in figure legends. After the transfer reaction, samples were cooled down on ice and mixed with 200 µl of assay buffer containing 37.5% sucrose. 480 µl of the mixture was transferred to 4.5 ml open-top centrifuge tubes for SW-60Ti rotor (Beckman coulter) and was overlaid with 1 ml of assay buffer containing 7.5% sucrose and then with 0.2 ml of assay buffer. The tubes were spun at 50,000 rpm [SW-60ti, 257,000×*g* (avg)] for 150 minutes at 4 °C. Upper 480 µl from the tubes (float fraction) were recovered to fresh tubes. NBD- and rhodamine fluorescence of the NBD-PE (in donor liposomes) and of the Rhodamine-PE (in acceptor liposomes) in samples before and after flotation were determined to check the contamination of donor liposomes and the recovery of acceptor liposomes in the float fraction, respectively. DOPA, DOPS, 17:0 PC and 17:0 PE amounts in the samples were determined by qMS. The amount of 17:0 PC or 17:0 PE in the float fraction is used to determine the recovery of acceptor liposomes or the contamination of donor liposomes in the fraction, respectively. After normalisation of recovery and contamination, the amounts transferred DOPA or DOPS were shown as a percentage of total input of them in the reaction.

**Cell fractionation and isolation of mitochondria**. Isolation of mitochondria from yeast cells was performed according to standard procedures[41]. Purity of mitochondria were assessed by SDS-PAGE and Western blotting for ER (Sec61) and for mitochondria (Cox2) markers according to standard procedures.

To isolate the total membrane fraction devoid of lipid droplet for lipid analysis of CL, 50 OD unit of logarithmically growing cells were treated by lyticase according to the standard protocol for isolation of mitochondria. The spheroplasts were resuspended carefully by pipetting with 1 ml of ice-cold sorbitol-phosphate buffer (1.2 M sorbitol, 20 mM kaliumphosphate buffer). After re-isolation of spheroplasts by centrifugation (1200 × *g*, 5 min, 4 °C), the pellets were resuspend in 1.3 ml of ice-cold hypotonic lysis buffer (10 mM Tris-HCl pH 7.4) and incubate for 5 min on ice. Unbroken cells were isolated by centrifugation (700 × *g*, 5 min, 4 °C) and the supernatant was recovered to a fresh tube. Again, unbroken cells were removed (700 × *g*, 5 min, 4 °C) and the supernatant was transferred to Beckmann's ultracentrifuge tubes. Total membrane fraction were isolated by ultracentrifugation (150,000 × *g*, 60 min, TLA-55). The supernatant (containing lipid droplet) was

removed and 500 µl of pure water was added gently to the tube without mixing. The tube was centrifuged (150,000 × g, 10 min, TLA-55/50,000 rpm) and the supernatant was removed carefully without disturbing the pellet. The pellet was resuspended in 100 µl of pure water and the protein concentration was determined by Bradford assay. Aliquots of 100 µg protein were kept at −80 °C and were subjected to lipid analysis by qMS.

**Quantitative mass spectrometry of phospholipids.** Mass-spectrometric analysis of phospholipids was performed essentially as described[41]. Briefly, lipids were extracted from samples in the presence of internal standards of major phospholipids (PC 17:0-14:1, PE 17:0-14:1, PI 17:0-14:1, PS 17:0-14:1, PA 17:0-14:1, PG 17:0-14:1 all from Avanti Polar Lipids) and CL (CL mix I, Avanti Polar Lipids LM-6003). Extraction was performed according to Bligh and Dyer with modifications. Final lipid samples were dissolved in 10 mM ammonium acetate in methanol and were splayed into a QTRAP 6500 triple quadrupole mass spectrometer (SCIEX) by nano-infusion splay device (TriVersa NanoMate with ESI-Chip type A, Advion). The quadrupoles Q1 and Q3 were operated at unit resolution. Phospholipid analysis was carried out in positive ion mode. PC analysis was carried out by scanning for precursors of $m/z$ 184 at a collision energy (CE) of 37 eV. PE, PI, PS, PG and PA measurements were performed by scanning for neutral losses of 141, 277, 185, 189 and 115 Da at CEs of 30, 30, 30, 25 and 25 eV, respectively. CL species were identified by scanning for precursors of the masses ($m/z$ 465.4, 467.4, 491.4, 493.4, 495.4, 505.5, 519.5, 521.5, 523.5, 535.5, 547.5, 549.5, 551.5, 573.5, 575.5, 577.5, 579.5, 601.5, 603.5, 605.5, 607.5, 631.5, 715.5 and 771.5 Da) corresponding DAG-$H_2O$ fragments as singly charged ions at CEs of 45 eV. Mass spectra were analysed by the LipidView Software Version 1.2 (SCIEX) for identification, correction of isotopic overlap and quantification of lipids. Correction of isotopic overlap in CL species was performed using a spreadsheet calculating and subtracting theoretical amounts of [M+2] and [M+4] isotopes of each CL species that are isobalic to other CL species. Lipid amounts (pmol) were corrected for response differences between internal standards and endogenous lipids.

**Identification of bound lipids to purified proteins.** Overall, 20 µg of protein were added with 1 µg of trypsin and left incubating overnight at 37 °C. The mixture was then dried in a SpeedVac concentrator and re-dissolved in 60% acetonitrile (ACN) by sonication for 10 min. The lipid samples were separated on a C18 column (Acclaim PepMap 100, C18, 75 µm × 15 cm; Thermo Scientific) by Dionex Ulti-Mate 3000 RSLC nano System, and then analysed by a hybrid LTQ-Orbitrap mass spectrometer (Thermo Scientific) via a dynamic nanospray source. The buffers and gradient used for separation of lipids are adapted from ref. [42]. Briefly, a binary buffer system was used with buffer A [ACN: $H_2O$ (60:40), 10 mM ammonium formate, 0.1% formic acid] and buffer B [IPA: ACN (90:10), 10 mM ammonium formate, 0.1% formic acid]. The phospholipids were separated with a gradient of 32 to 99% buffer B at a flow rate of 300 nl/min over 30 min and the column was kept at 40 °C. Typical electrospray conditions were a spray voltage of 1.8 kV and capillary temperature of 175 °C. The LTQ-Orbitrap XL was operated in negative ion mode and in data-dependent acquisition with one MS scan followed by five MS/MS scans. Survey full-scan MS spectra were acquired in the Orbitrap within the mass range from $m/z$ 350 to 2000 at a resolution of 60,000. Collision-induced dissociation (CID) fragmentation in the linear ion trap was performed for the five most intense ions at the following settings: automatic gain control target of 30,000, normalised collision energy of 38% at an activation of $q = 0.25$ and activation time of 30 ms.

**Molecular simulations.** Coarse-grained (CG) molecular dynamics simulations were performed using the gromacs 5 software (www.gromacs.org) with GPU acceleration. Atomistic monomeric models of PRELID3b and PRELID1 were generated using Modeller 9v9 (ref. [43]) using the monomeric crystal structure of PRELID3a (PDB id 4xzv[19]) as a template to orientate the C-terminal helix[19]. 10 models of each were built of which the lowest scoring (molpdf) was used for the CG conversion. CG models of PRELID1, PRELID3b and TRIAP1 were generated using the martinize.py script, with the MARTINI 2.2 force field[44], using an elastic network with upper and lower cut offs of 1 and 0.6 nm and a force constant of 500 kJ nm. The starting position of the lipid in the cavity was based on alignment with the X-ray density observed in the map for PRELID3b, and by alignment with Ups1 for PA[20]. POPS was used for PRELID3b and POPA for PRELID1. Ups1–Mdm35 complexes were built using PDB id 4xiz[21].

Model mitochondrial membranes were generated by self-assembly. The compositions were chosen to reflect those used in the lipid transfer assays presented here. For PRELID3b this was POPC:POPE:CDL:POPS 50:40:5:5 and for PRELID1/Ups1 POPC:POPE:CDL:POPA 50:40:5:5. The PRELI–TRIAP1/Ups1–Mdm35 complexes were positioned 10 nm from the membrane and randomly orientated in $x$, $y$ and $z$ dimensions to generate three independent initial configurations. Simulations were performed as described elsewhere[45]. Briefly, simulations were performed at 310 K using the V-rescale algorithm and 1 tau using the Berendsen barostat with semiisotropic coupling[46]. 50,000 steps of equilibration were performed with the protein backbone positionally restrained. Production simulations were 5 µs using a 20 fs timestep. Visualisation used Pymol (www.pymol.org) and VMD[47]. The lipid contact analysis script was provided by Heidi

Koldsoe (Schrodinger Llc). Lipid contacts were calculated between each residue of the protein and the stated particle type of a given lipid using a distance cutoff of 0.6 nm to define a contact.

**Quantification and statistical analysis.** Quantitative data are presented as arithmetic means ± standard error (SEM). The statistical significance in related figures was assessed using two-tailed Student's $t$-test. A $p$ values and $n$ in column plots from Student's $t$-test were specified in corresponding figure legend.

**Code availability.** The script used to analyse protein–lipid interactions has been previously reported[48], and is available from the corresponding authors upon request and with the permission of Heidi Koldsø.

## Data availability
The structure data for TRIAP1–PRELID3b, TRIAP1–PRELID1 and TRIAP1–PRELID1 (K58V) complexes have been deposited in the Protein Data Bank (with PDB ID codes: 6I4Y, 6I3V, 6I3Y). Other data are available from the corresponding authors upon reasonable request. The source data underlying Figs. 2b and 5b are provided as Supplementary Data 1. Uncropped gel images used to generate panels in Supplementary Figure 2C, 3A, 4A, 5C, 6A, 6C and 6H are provided in Supplementary Figure 11. The source data of Figs. 3c, 4b, 6d, 6f and Supplementary Figure 2D, 3D, 3E, 4B, 6B, 6D, 6E and 6H are provided as a Source Data file.

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

## Acknowledgements

We are particularly grateful and thankful to Dr Marc Morgan as the Protein Crystallography Facility Manager for assistance in data processing. We thank Gudrun Zimmer for excellent technical support and Philipp Lampe for support in cell-free synthesis. We also thank the support of several undergraduate project students from Imperial College. This work was supported by funds from the Medical Research Council (MR/M019403/1) to S.M. and grants of the Deutsche Forschungsgemeinschaft to T.T. (TA1132/2-1) and T.L. (LA918/14-1).

## Author contributions

T.T. and K.S. performed the genetic screens, molecular biology and functional assays experiments. X.M. and J.B. performed molecular biology, protein production and crystallisation experiments. X.M. and J.B. carried out the diffraction data collection and structure determination. S.L.R. performed the molecular dynamics simulations. T.T. performed quantitative lipidomics. D.S.C., D.W. and C.V.R. carried out mass spectrometry. X.M., J.-L.B., T.T., S.L.R., T.L. and S.M. contributed to the experimental design, data analysis and discussion and wrote the manuscript. T.L. and S.M. conceived the project and secured funding.

## Additional information

**Competing interests:** The authors declare no competing interests.

