## [Peer Review File · Nature Communications]

Reviewers' comments:

Reviewer #1 (Remarks to the Author):

Ups1/PRELID1-Mdm35/TRIAP1 and Ups2/PRELID3b-Mdm35/TRIAP1 mediate phosphatidic acid (PA) and phosphatidylserine (PS) transfer, respectively, between the mitochondrial outer membrane (OM) and inner membrane (IM). Previous studies on these proteins showed that hydrophobic phospholipid molecules are more or less shielded from the aqueous environments for lipid transfer between membranes, yet how different lipid molecules are recognized and selected for lipid transfer by lipid transfer proteins are still a mystery. In the present study, Miliara et al. tackled this problem, i.e. still elusive structural basis of the substrate specificity for lipid transfer proteins within mitochondria, PA-specific Ups1/PRELID1 and PS-specific Ups2/PRELID3b. The authors reported here the new crystal structures of those two proteins complexed with TRIAP1. The structures themselves did not add so much to the present understanding of the lipid recognition by these proteins. However, by using a yeast reverse genetic screen, the authors identified mutations in Ups1 that complemented the defects due to UPS2 deletion. The revealed mutations were mapped onto the region facing to the lipid binding cavity and near the Ω loop functioning as a lid for the lipid binding cavity in the Ups1 structure. Ups1 variants carrying these mutations showed broadened lipid specificity rather than converting it from PA to PS *in vitro*. Screen for Ups2 mutants in the other way around showed the importance of the Ω loop regions for substrate specificity, and indeed the Ups2 variant carrying the Ω loop of Ups1 was able to transfer PA. Thus, the yeast screen revealed the hotspots for the determinants of lipid substrate specificity. The manuscript is well written, and the experiments are technically sound. The present work provides important insight into the previously elusive structural basis for the substrate specificity for lipid-transfer proteins. I have the following suggestions to increase the impact and strength of the manuscript.

(1) K58V mutation

It is easy to understand that enlarging the lipid binding cavity by replacing K58 with V58 with a shorter side chain allows binding of a larger substrate, PS, to the cavity in addition to a smaller substrate PA. Nevertheless, experimental support for this interpretation is not strong since electron density of PS in the K58V mutant is extremely low. Fo-Fc difference Fourier map at 3σ instead of that at 1σ had better be presented to indicate the substrate position. In Fig. 4C and D, it is not appropriate to refer to the data as a PDB file of the K58V mutant alone, not the complex with PS for detailed discussion on the mutant-PS complex, which hampers the validation of the structural basis of the interpretation. The authors should deposit the PDB file for the complex with the coordinate of "the modeled PS" for discussion. The authors stated that the structure resolution was 2.7Å, but the statistics showed that it was 2.98Å.

(2) Loop mutant

An intellectually interesting point is that Ups2 can accommodate a larger substrate PS more efficiently than a smaller substrate PA. The authors identified Ups2 mutants with increased PA binding, but retaining PS binding. Those mutation points were enriched in the Ω loop, suggesting its role in PS selection as a substrate, and indeed Ups2 with the replaced the Ω loop with the Ups1 Ω loop gained a PA transfer activity. The role of the Ω loop in the substrate specificity was discussed on the basis of MD simulation, attributing to the hydrophobicity difference between the two proteins. However, the MD simulation was not strong enough to persuade readers of their model that PA, not PS, was released from the complex with only partial burial of the Ups protein. More experimental evidence is required to validate this discussion. It would be also informative to test the alternative loop swap mutant of Ups1 containing the Ups2 Ω loop. In addition, Fig. 7 and the movie are not reader-friendly. The color of the loop region had better be changed in Fig. 7 and the movie to make the loop more visible than other regions. The movie for PRELID3b-PS, in addition to PRELID1-PA, would help understanding by readers, as well.

(3) Lipid transfer analyses

Lipid transfer specificity of the mutants were analyzed only by in vitro lipid transfer assays. The recovery of the decreased cardiolipin or phosphatidylethanolamine levels by the Ups1 variants with increased PS transfer activity or by the Ups2 variants with increased PA transfer activity, respectively, in the *ups1Δups2Δ* strain should be tested. In vitro lipid transfer assays employed PS or PA containing the NBD group introduced probably in the acyl chains (this should be described in the manuscript, instead of referring to the previous papers). The hydrophilic NBD group may affect extractability of the lipid from the membrane, which may affect substrate lipid specificity of the Ups1 or Ups2 variants. The authors had therefore better use mass spectrometry in lipid transfer assays in vitro, in addition to the fluorescence-based lipid transfer assays.

(4) Other points

In Fig. 2C and Fig. 5C, the expression levels of the Ups mutants should be shown.

In the reverse genetic screen, the method should be described more precisely, i.e. how many colonies were picked up, and how plasmids were recovered from growing colonies etc.

Reviewer #2 (Remarks to the Author):

This is a straightforward paper that uses state-of-the-art approaches to advance understanding of the mechanism by which the Ups/Mdm35 (PRELI-TRIAP) complexes move lipids between the outer and inner membrane of mitochondria. New crystal structures complete the primary gallery of structures for members of this family, and genetics approaches identify key residues involved in selectivity (PA versus PS). Some validation of the lipid ejection mechanism is attempted through molecular dynamics simulations.

1. There are errors in language throughout the manuscript, and in some cases typographical errors in the labeling of figures. Examples of the latter: activity is spelled incorrectly in Fig 3C; transported is spelled incorrectly in Fig. 4A.

2. The activity assays include an offset that the authors do not comment on. For example, Fig. 3A,B. Why do the traces start at different levels of fluorescence? The quenched NBD-PS liposomes are the same for all assays, the only difference between the individual traces being the protein variant being supplied to catalyze transfer.

3. The molecular dynamics simulations are cursory. There is very little (if any) quantification on offer. This section must be fleshed out to provide more details of the lipid ejection mechanism - identify participating residues in the protein, indicate a time frame, etc. The simulations are performed on the coarse-grained level where sampling of unstructured regions, such as loops, is suspect at best. It is dangerous to draw conclusions about the omega loop dynamics from these simulations. It would be advisable to transform some of the structures into all atom representation and run all-atom MD simulations. Specific points on the MD simulations are below:

Page 15. "The complex contacts the membrane several times before converging upon the stable binding mode, which involves the conformational change and insertion of the omega loop and part of the C-terminal helix in the bilayer."

Questions: How is the stability of the binding mode defined? What criteria were used? What kind of conformational change occurs? No details are presented either in the text or in the figures.

Page 16, "Interestingly, we found that PA was released from the complex with only partial burial of the protein, limited to the omega loop and one end of the adjacent C-terminal helix, which are deeply buried into the hydrophobic region of the membrane. PS release, on the other hand, involved a more extensive overall interface of the protein surface on the membrane at the point of lipid release"

Questions: How many release events were sampled for PA and how many for PS? What can be

said about reproducibility of the binding mode during the release for these two lipids? How are the authors certain that PS is not released under the conditions that PA is and vice versa?

Reviewer #3 (Remarks to the Author):

The manuscript by Miliara et al. reports a study of mitochondrial phospholipid transfer proteins. It is an interdisciplinary study, including genetic approaches, protein crystallography and molecular dynamics (MD) simulations. Miliara et al. have determined new crystal structures of two lipid transfer proteins of different specificity, and used genetic screens to identify specific residues as well as flexible loop regions involved in the lipid specificity. The manuscript is well written. I am not an expert on the lipid transfer proteins or genetic screens, but the work appears to be very thorough. I have been asked specifically to evaluate the MD simulations presented in the manuscript, which is what I have focused on. Overall, things seem fine. My main question is:

- How consistent are the results from the simulation repeats? It is not clear from the results section whether more than one repeat has been performed, but the method section indicates more than one repeat. I.e. is it consistent between repeats that PA is released from a partly buried protein, while PS release requires a larger protein-membrane interface? From the method section it seems as if three repeats were performed – all with same results? Were all simulations included in calculating the lipid density and “release angle”?

A few minor comments and suggestions:

- Perhaps include the chemical structure of PS and PA to help readers who don't remember their lipids by heart
- When presenting the results from the MD simulations, perhaps mention that they are performed at the coarse-grained (CG) level of detail.
- P. 18, “and” missing – F133, M135 should probably be F133 and M135
- Legend for Fig. 1: is RMSD here backbone RMSD as mentioned on p12?
- Figure 4: Potentially add labels to help the reader identify K58V and loop 2.
- Figure 7: labels A-C missing. Phosphate groups do not seem transparent in panel A (despite what it says in the legend), but they are fine as they are.
- MD methods: Ideally include version number of Modeller for reproducibility. Was more than one model generated? If yes, how was the best model chosen?

Point-to-point response to reviewers

Reviewer #1 (Remarks to the Author):

Ups1/PRELID1-Mdm35/TRIAP1 and Ups2/PRELID3b-Mdm35/TRIAP1 mediate phosphatidic acid (PA) and phosphatidylserine (PS) transfer, respectively, between the mitochondrial outer membrane (OM) and inner membrane (IM). Previous studies on these proteins showed that hydrophobic phospholipid molecules are more or less shielded from the aqueous environments for lipid transfer between membranes, yet how different lipid molecules are recognized and selected for lipid transfer by lipid transfer proteins are still a mystery. In the present study, Miliara et al. tackled this problem, i.e. still elusive structural basis of the substrate specificity for lipid transfer proteins within mitochondria, PA-specific Ups1/PRELID1 and PS-specific Ups2/PRELID3b. The authors reported here the new crystal structures of those two proteins complexed with TRIAP1. The structures themselves did not add so much to the present understanding of the lipid recognition by these proteins. However, by using a yeast reverse genetic screen, the authors identified mutations in Ups1 that complemented the defects due to UPS2 deletion. The revealed mutations were mapped onto the region facing to the lipid binding cavity and near the Ω loop functioning as a lid for the lipid binding cavity in the Ups1 structure. Ups1 variants carrying these mutations showed broadened lipid specificity rather than converting it from PA to PS in vitro. Screen for Ups2 mutants in the other way around showed the importance of the Ω loop regions for substrate specificity, and indeed the Ups2 variant carrying the Ω loop of Ups1 was able to transfer PA. Thus, the yeast screen revealed the hotspots for the determinants of lipid substrate specificity. The manuscript is well written, and the experiments are technically sound. The present work provides important insight into the previously elusive structural basis for the substrate specificity for lipid-transfer proteins. I have the following suggestions to increase the impact and strength of the manuscript.

(1) K58V mutation

It is easy to understand that enlarging the lipid binding cavity by replacing K58 with V58 with a shorter side chain allows binding of a larger substrate, PS, to the cavity in addition to a smaller substrate PA. Nevertheless, experimental support for this interpretation is not strong since electron density of PS in the K58V mutant is extremely low. Fo-Fc difference Fourier map at 3σ instead of that at 1σ had better be presented to indicate the substrate position. In Fig. 4C and D, it is not appropriate to refer to the data as a PDB file of the K58V mutant alone, not the complex with PS for detailed discussion on the mutant-PS complex, which hampers the validation of the structural basis of the interpretation. The authors should deposit the PDB file for the complex with the coordinate of "the modeled PS" for discussion. The authors stated that the structure resolution was 2.7Å, but the statics showed that it was 2.98Å.

We have submitted the PS-bound form of PRELID1-K58V to the PDB under code 6I3Y. The occupancy of ligand PS is set to 0.3 for final refinement, which is in accordance with mass spectrometry and lipidomics analyses confirming 30% ligand exchange (see supplementary Fig. S4D). Figure 4 in the manuscript has been updated with final refined model including PS.

The contour levels σ of the Fo-Fc map surrounding the PS molecule in Figure 4 C, were set to 2.0 originally but mistyped as 1.0 in the text. The supplementary figure below shows the map with σ -levels 1.0, 2.0 and 3.0. The figure in the manuscript has been updated with the map set to the suggested σ -level 3.0.

Typographical error relating to the resolution as 2.98Å has now been corrected in the text.

To further substantiate the importance of K58 in substrate recognition experimentally, we now show that Ups1-K58V is able to transport a non-fluorescent, endogenous PS (DOPS) in the presence of PA (DOPA) in the membrane, ensuring the benefit of the K to V exchange for PS transport (new Fig. S3EF). Moreover, new assessment of MD simulation data supports the role of the K58 in PA recognition, while the residue is constantly associated with the phosphate group of PA in the cavity (new Fig. 7D).

(2) Loop mutant

An intellectually interesting point is that Ups2 can accommodate a larger substrate PS more efficiently than a smaller substrate PA. The authors identified Ups2 mutants with increased PA binding, but retaining PS binding. Those mutation points were enriched in the Ω loop, suggesting its role in PS selection as a substrate, and indeed Ups2 with the replaced the Ω loop with the Ups1 Ω loop gained a PA transfer activity. The role of the Ω loop in the substrate specificity was discussed on the basis of MD simulation, attributing to the hydrophobicity difference between the two proteins. However, the MD simulation was not strong enough to persuade readers of their model that PA, not PS, was released from the complex with only partial burial of the Ups protein. More experimental evidence is required to validate this discussion.

As outlined before, we now show that the loop-swap mutant of Ups2 is able to transport a non-fluorescent, endogenous PA (DOPA) in the presence of PS (DOPS), ensuring the benefit of the hydrophilic Ω loop of Ups1 for PA transfer (new Fig. S6EF).

We have performed additional MD simulations including the PRELID3b variant carrying the Ω loop of PRELID1 and Ups1 (new Figs. 7, S9 and Table S6). Whereas all molecules carrying Ups1/PRELID1-type hydrophilic Ω loop bind the model membrane in an “upright” position during substrate release, PRELID3b harboring a hydrophobic Ω loop form a more extensive binding surface. The PRELID3b-loop swap variant behaves substantially different from PRELID3b, and keeps “upright” position. The positioning on the membrane in MD simulation correlate nicely with the substrate preference of these molecules, which are now validated

both in vitro and in vivo. These new indications support our proposal that the membrane binding mode is a determinant of substrate selection and that the properties of the Ω loop adjust the position suitable for a substrate.

It would be also informative to test the alternative loop swap mutant of Ups1 containing the Ups2 Ω loop.

As the reviewer suggested, we tested lipid transfer activity of the Ups1 loop-swap variants (with Ups2's Ω loop, corresponding either to amino acids 60-71 or amino acids 58-71 including 58V) in vitro. These variants showed very low transfer activities for both PA and PS (new Figure S6GH). This result indicates that the positively charged Ω loop of Ups1 is essential for its PA transfer activity. The result also suggests that the hydrophobic loop from Ups2 is not sufficient to allow Ups1 to transfer PS efficiently and an additional segment is required, which may function cooperatively with the Ω loop. We assume that the introduction of the hydrophobic Ω loop to Ups1, a mismatch of membrane binding positions and substrate handover process in the cavity prevents efficient capture of either PA or PS. This indication fits perfectly to the result from the forward genetic screen for Ups1 mutants (Fig. 2B). In the screen, almost no mutation could be identified in the Ω loop, indicating that a simple amino acid exchange in the loop is not sufficient to allow Ups1 to transfer PS. We have added these results and discuss it in the text (Fig. S6H, p20L13-16).

In addition, Fig. 7 and the movie are not reader-friendly. The color of the loop region had better be changed in Fig. 7 and the movie to make the loop more visible than other regions. The movie for PRELID3b-PS, in addition to PRELID1-PA, would help understanding by readers, as well.

We have completely revised Figure 7 and the movies to highlight the Ω loop region more clearly, and have included a movie of PRELID1-PA as well as PRELID3b-PS (new Figs. 7, S7, movie S1 and S2).

(3) Lipid transfer analyses

*Lipid transfer specificity of the mutants were analyzed only by in vitro lipid transfer assays. The recovery of the decreased cardiolipin or phosphatidylethanolamine levels by the Ups1 variants with increased PS transfer activity or by the Ups2 variants with increased PA transfer activity, respectively, in the *ups1 Δ ups2 Δ* strain should be tested.*

We agree that the analysis of CL and PE level is a good strategy to assess PA and PS transfer activity in vivo. However, the *ups1 Δ ups2 Δ* strain is not a suitable host strain, because PE and CL metabolism are interdependent. We have found previously that the additional deletion of *UPS2* in *ups1 Δ* restores CL levels by activation of an uncharacterized, alternative lipid transport route, allowing CL synthesis likely from another precursor lipid (Connerth et al., 2012). Therefore, we have measured PE levels in mitochondria of *ups2 Δ* cells expressing Ups1 mutants (K58V, E108D and K58VE108D, Fig. S2C) and CL levels in *ups1 Δ* cells expressing either Ups1 (positive control), Ups2 or the Ups2 loop-swap variant (new Fig. 6F). The expression of the Ups1 variants in *ups2 Δ* cells or of the Ups2 loop-swap variant in *ups1 Δ* cells

significantly increased mitochondrial PE and CL levels, respectively, demonstrating enhanced PS and PA transfer activity in vivo.

In vitro lipid transfer assays employed PS or PA containing the NBD group introduced probably in the acyl chains (this should be described in the manuscript, instead of referring to the previous papers). The hydrophilic NBD group may affect extractability of the lipid from the membrane, which may affect substrate lipid specificity of the Ups1 or Ups2 variants. The authors had therefore better use mass spectrometry in lipid transfer assays in vitro, in addition to the fluorescence-based lipid transfer assays.

We have now tested PA/PS transport activities of the mutant variants of Ups1 or Ups2 using non-fluorescent PA (DOPA) and PS (DOPS) combined with mass spectrometry. These experiments confirmed our findings using NBD labelled lipids that we now describe in more detail in the method section. To mimic physiological conditions, we added both DOPA and DOPS in the donor liposomes in these experiments. The loop-swap mutant of Ups2 is able to transfer DOPA in the presence of DOPS, illustrating that the hydrophilic loop from Ups1 promotes PA transport (Fig. S6EF). The Ups1^{K58V} and Ups1^{K58VE108D} are able to transport DOPS in the presence of DOPA, further supporting the beneficial effect of the K58V and E108D mutations for PS transport (Fig. S3EF).

(4) Other points

In Fig. 2C and Fig. 5C, the expression levels of the Ups mutants should be shown.

We have monitored the expression levels of Ups1 and Ups2 mutant variants by immunoblotting (new Figs. S2D, S5C). The analysis of cells expressing Ups2 variants revealed largely similar protein levels as observed in wild type cells revealed (Fig. S5C). When comparing the expression level of Ups1 variants, we found that proteins harboring the K58V mutation accumulated at higher levels in mitochondria (Fig. S2D). While this may contribute to the observed increase in PA transfer activity in vivo, it does not affect the main conclusion on the role of K58 in determining lipid specificity, as the transfer specificity was assessed in vitro using purified proteins present at the same concentration.

In the reverse genetic screen, the method should be described more precisely, i.e. how many colonies were picked up, and how plasmids were recovered from growing colonies etc.

We have added the respective information in the method section.

Reviewer #2 (Remarks to the Author):

This is a straightforward paper that uses state-of-the-art approaches to advance understanding of the mechanism by which the Ups/Mdm35 (PRELI-TRIAP) complexes move lipids between the outer and inner membrane of mitochondria. New crystal structures complete the primary gallery of structures for members of this family, and genetics approaches identify key residues involved in selectivity (PA versus PS). Some validation of the lipid ejection mechanism is attempted through molecular dynamics simulations.

1. There are errors in language throughout the manuscript, and in some cases typographical errors in the labeling of figures. Examples of the latter: activity is spelled incorrectly in Fig 3C; transported is spelled incorrectly in Fig. 4A.

We have carefully checked the text and corrected errors.

2. The activity assays include an offset that the authors do not comment on. For example, Fig. 3A,B. Why do the traces start at different levels of fluorescence? The quenched NBD-PS liposomes are the same for all assays, the only difference between the individual traces being the protein variant being supplied to catalyze transfer.

The offset appears because our plate reader does not allow us to monitor fluorescence in the first ~10 seconds after starting the transfer reaction by the addition of acceptor liposomes. We prefer not to correct for this offset and show raw data because the offset itself is informative and reflects the transfer activity. To avoid the offset, we have tried to slow down the reaction by using lower protein concentration in the assay. However, the linearity between activity and concentration was lost under these conditions, suggesting an additional rate-limiting step at low protein concentrations.

3. The molecular dynamics simulations are cursory. There is very little (if any) quantification on offer. This section must be fleshed out to provide more details of the lipid ejection mechanism - identify participating residues in the protein, indicate a time frame, etc.

We have expanded the set of simulations from 2 to 5, and include a more detailed analysis of all of the sets of trajectories. The new simulations are of

- 1) PRELID1 K58V mutant with PS,
- 2) PRELID3b-loop swap (variant carrying the Ω loop of PRELID1) with PA,
- 3) Ups1 with PA.

Details of all simulations are summarised in a new Table S6. We also present further analysis in new figures (new Fig. 7, Fig. S7-9), including a quantitative analysis of the contacts between the lipid and protein during the steps of lipid release, and have updated the discussion accordingly. Importantly, the new MD simulations revealed a two-step release mode of lipids from Ups/PRELID proteins and identified residues that were identified in the genetic screens as forming key contacts with the lipid substrate within the primary site (Fig. 7).

The simulations are performed on the coarse-grained level where sampling of unstructured regions, such as loops, is suspect at best. It is dangerous to draw conclusions about the omega loop dynamics from these simulations. It would be advisable to transform some of the structures into all atom representation and run all-atom MD simulations.

We thank the reviewer for this suggestion. We chose to use coarse grain (CG) level of resolution to monitor the membrane binding of the complexes from solution, which requires relatively long timescales that would not be accessible by atomic simulation. We recognise that CG methodology has limitations including the fixed secondary structure, however the omega loop contains a helical segment and is not completely disordered. We feel that a full attempt to fully characterise the conformations of the disordered parts of the omega loop by all-atom simulation, although valuable, is beyond the scope of this manuscript. We have added a comment to the discussion on the limitations of CG approaches and potential of using e.g. enhanced sampling methods to characterise loop regions (p.20, L17).

Specific points on the MD simulations are below:

Page 15. "The complex contacts the membrane several times before converging upon the stable binding mode, which involves the conformational change and insertion of the omega loop and part of the C-terminal helix in the bilayer."

Questions: How is the stability of the binding mode defined? What criteria were used? What kind of conformational change occurs? No details are presented either in the text or in the figures.

We now highlight the residue-by-residue interactions with the membrane headgroups per simulation, to show that the same regions bind and, once bound, remain so for the duration of each simulation (Figure S8A). We have carried out more involved analyses and include a new figures (Figure 7, S7, S8A) with the aim to provide the reader with a clearer picture of the stages of membrane binding.

Page 16, "Interestingly, we found that PA was released from the complex with only partial burial of the protein, limited to the omega loop and one end of the adjacent C-terminal helix, which are deeply buried into the hydrophobic region of the membrane. PS release, on the other hand, involved a more extensive overall interface of the protein surface on the membrane at the point of lipid release"

Questions: How many release events were sampled for PA and how many for PS? What can be said about reproducibility of the binding mode during the release for these two lipids? How are the authors certain that PS is not released under the conditions that PA is and vice versa?

In order to address this question we have expanded the set of simulations to include the PRELID1 K58V mutant with PS, and a model of the PRELID3b loopswap mutant with PA. We also included a set of Ups1-PA simulations to provide further evidence of the membrane bound orientation. The details of each are included in Table S6. PRELID1 K58V was able to release PS in an upright configuration, and therefore it appears the orientation on the membrane is defined by the properties of the PRELI domain rather than by lipid type.

Reviewer #3 (Remarks to the Author):

The manuscript by Miliara et al. reports a study of mitochondrial phospholipid transfer proteins. It is an interdisciplinary study, including genetic approaches, protein crystallography and molecular dynamics (MD) simulations. Miliara et al. have determined new crystal structures of two lipid transfer proteins of different specificity, and used genetic screens to identify specific residues as well as flexible loop regions involved in the lipid specificity. The manuscript is well written. I am not an expert on the lipid transfer proteins or genetic screens, but the work appears to be very thorough. I have been asked specifically to evaluate the MD simulations presented in the manuscript, which is what I have focused on. Overall, things seem fine. My main question is:

How consistent are the results from the simulation repeats? It is not clear from the results section whether more than one repeat has been performed, but the method section indicates more than one repeat. I.e. is it consistent between repeats that PA is released from a partly buried protein, while PS release requires a larger protein-membrane interface? From the method section it seems as if three repeats were performed – all with same results? Were all simulations included in calculating the lipid density and “release angle”?

As discussed above, details of simulations performed have been added to Table S6. We also expanded the number of simulation sets. The stochastic nature of membrane binding, lipid release and subsequent interaction of lipids with the secondary binding site is highlighted by Table S6 and Figure S9. However, PRELID3b consistently adopted the flat orientation, whilst PRELID1 was upright for all lipid release events. We have removed the use of ‘release angle’ as it is perhaps oversimplified and instead use direct measures of residue-by-residue contacts. We now make clear in the captions whether data is from single representative simulations or averaged over the 3 repeats.

A few minor comments and suggestions:

- Perhaps include the chemical structure of PS and PA to help readers who don't remember their lipids by heart

This has been incorporated into Fig S7B.

- When presenting the results from the MD simulations, perhaps mention that they are performed at the coarse-grained (CG) level of detail.

This has been added to the text.

- P. 18, “and” missing – F133, M135 should probably be F133 and M135

Thank you for the correction. We have corrected the error.

- Legend for Fig. 1: is RMSD here backbone RMSD as mentioned on p12?

This is correct. We also mention this now in the legend for Fig. 1.

- *Figure 4: Potentially add labels to help the reader identify K58V and loop 2.*

We have added annotations to loop 2 in Fig. 4C.

- *Figure 7: labels A-C missing. Phosphate groups do not seem transparent in panel A (despite what it says in the legend), but they are fine as they are.*

We have extended our MD simulations and show the results in new figures 7 and Figure S7-S9.

- *MD methods: Ideally include version number of Modeller for reproducibility. Was more than one model generated? If yes, how was the best model chosen?*

This has been added to the methods.

REVIEWERS' COMMENTS:

Reviewer #1 (Remarks to the Author):

This is a revised version of the manuscript that was previously submitted to Nature Communications. The authors addressed most of my concerns by performing several new experiments and the manuscript was improved significantly. The results of MD simulation are now presented in an improved way that readers can easily understand the points the authors want to insist. This is an important work that needs to be published urgently.

Reviewer #2 (Remarks to the Author):

The authors have systematically and satisfactorily responded to the previous comments. I have only one further point concerning the offset in the activity assays - the authors explained this in their 'rebuttal' but they should include these comments in the methods section or even in the main text.

Reviewer #3 (Remarks to the Author):

The manuscript by Miliari et al. has been much improved upon the revisions. I have focused on the computational part, and the new MD simulations and the much more rigorous analysis and description of results have made this section much stronger. My only remaining question is: how are "contacts" defined in the heatmaps in Figure 7?

I think the criteria for defining a contact should be outlined in the methods or in the figure text.

Other than that, I recommend publication.